

**Effects of Sea Animal Colonization on the Coupling between Dynamics and**
**Activity of Soil Ammonia-oxidizing Bacteria and Archaea in Maritime Antarctica**
**Qing Wang [1], Renbin Zhu[1], Yanling Zheng[2], Tao Bao[1], Lijun Hou[2]**
[1]Anhui Province Key Laboratory of Polar Environment and Global Change, School of Earth and
Space Sciences, University of Science and Technology of China, Hefei 230026, P.R China
[2]State Key Laboratory of Estuarine and Coastal Research, East China Normal University,
Shanghai 200062, P. R China
Corresponding author: Renbin Zhu (zhurb@ustc.edu.cn) or Lijun Hou
(ljhou@sklec.ecnu.edu.cn)





## Abstract

The colonization of a large number of sea animal, including penguins and seals, plays an important role in the nitrogen cycle of the tundra ecosystem in coastal Antarctica. However, little is known about the effects of sea animal colonization on ammonia-oxidizing archaea (AOA) and bacteria (AOB) communities involved in nitrogen transformations. In this study, we chose active seal colony tundra soils (STS), penguin colony soils (PTS), adjacent penguin-lacking tundra soils (PLS), tundra marsh soils (MS), and background tundra soils (BS), to investigate the effects of sea animal colonization on the abundance, activity, and diversity of AOA and AOB in maritime Antarctica. Results indicated that AOB dominated over AOA in PTS, STS, and PLS; whereas AOB and AOA abundances were similar in MS and BS. Penguin or seal activities increases the abundance of soil AOB $amoA$ genes, but reduced the abundance of AOA $amoA$ genes, leading to very large ratios ($1.5 \times 10^2$ to $3.2 \times 10^4$) of AOB to AOA $amoA$ copy numbers. Ammonia oxidation rates were significantly higher ($P = 0.02$) in STS and PTS than in PLS, MS, and BS, and were significantly positively correlated ($P < 0.001$) with AOB $amoA$ gene abundance suggesting that AOB are more important in the nitrification in animal colony soils. Sequence analysis for gene clones showed that AOA and AOB in tundra soils were from the $Nitrosospira$ and $Nitrososphaera$ lineages, respectively. Seal or penguin activities led to the predominant existence of AOA phylotypes related to $Nitrososphaera$ cluster I and AOB phylotypes related to $Nitrosospira$ clusters I and II, but very low relative abundances in AOA phylotypes related to cluster II, and AOB phylotypes related to cluster III and IV. The differences in AOB and AOA community structures were closely related to soil biogeochemical processes under the disturbance of penguin or seal activities: soil C:N alteration and sufficient input of $NH_4^+$–N and phosphorus from animal



excrements. The results provide insights into the mechanisms how microbes drive nitrification in
maritime Antarctica.
**Keywords:** Antarctic tundra, AOA, AOB, Marine animals, Nitrification, Nitrogen deposition



## 1 Introduction

Nitrification, the oxidation of ammonia into nitrate through nitrite, plays a pivotal role in the
global biogeochemical cycle for nitrogen (Nunes-Alves, 2016). As the first and rate-limiting step
of nitrification, ammonia oxidation (the aerobic oxidation of ammonia into nitrite) is performed
by phylogenetically and physiologically distinct groups of ammonia oxidizing archaea (AOA)
and ammonia oxidizing bacteria (AOB) (Könneke et al., 2005; Wang et al., 2015). Only recently
were the comammox, which directly oxidize ammonia to nitrate on their own, identified in the
bacterial genus *Nitrospira* (Daims et al., 2015; Kessel et al., 2015). The AOA and AOB have been
investigated using the *amoA* gene as a functional marker in a wide variety of environments,
including soils (Di et al., 2009; Gubry-Rangin et al., 2017; Leininger et al., 2006; Ouyang et al.,
2016; Shen et al., 2012), sediments (Li et al., 2015; Zheng et al., 2013), estuaries (Dang et al.,
2008; Mosier et al., 2008; Santoro et al., 2011), oxic and suboxic marine layers (Baker et al., 2012;
Bouskill et al., 2012), plateau permafrost (Zhang et al., 2009; Zhao et al., 2017), and in sub-arctic
and arctic soil (Alves et al., 2013; Daebeler et al., 2017). Results indicated that the relative
abundance and functional importance of AOA *vs*. AOB vary greatly in natural ecosystems.
Environmental drivers, including substrate concentration, oxygen availability, pH, and salinity,
might be responsible for the different AOA and AOB abundances and distribution (Alves et al.,
2013; Bouskill et al., 2012; Le Roux et al., 2008; Wang et al., 2015). The abundance, diversity,
and activity of ammonia-oxidizers have been explored in tundra soils of Antarctic Peninsula (Jung
et al., 2011; Yergeau et al., 2007), the Antarctic Dry Valleys (Ayton et al., 2010; Magalhães et al.,
2014; Richter et al., 2014), and in the Antarctic coastal waters (Kalanetra et al., 2009; Tolar et al.,
2016). However, there has been limited research about the abundance and diversity of microbes





and genes involved in the nitrogen cycle in the remote Antarctic terrestrial ecosystems. There is
still a large gap in our understanding of factors that control AOA *versus* AOB prominence, and
the relationships between nitrification rates and ammonia-oxidizer dynamics need to be explored
in the Antarctic.

In maritime Antarctica, a large number of sea animals, such as penguins or seals, settle on

some coastal ice-free tundra patches. Tundra vegetation including mosses, lichens, and algae,
penguin colonies, and their interactions, form a special ornithogenic tundra ecosystem (Tatur et
al., 1997). The soil biogeochemistry of an ornithogenic tundra ecosystem has become a research
hotspot under the penguin-activity disturbance (Otero et al., 2018; Riddick et al., 2012; Simas et
al., 2007; Zhu et al., 2013, 2014). Previous studies indicated that sea animals significantly affect
the tundra N and P cycles (Lindeboom et al., 1984; Simas et al., 2007; Zhu et al., 2011), and the
total N and P excreted by seabird breeders and chicks are 470 Gg N $yr^{-1}$ and 79 Gg P $yr^{-1}$ in
Antarctica and the Southern Ocean, accounting for 80% of the N and P from total global seabird
excreta (Otero et al., 2018). Uric acid is the dominant N compound in penguin guano, and during
its mineralization, different N forms, such as $NH_3$, $NH_4^+$, and $NO_3^-$, can be produced via
ammonification, nitrification, and deposition, following the changes in soil pH and the C:N ratio
(Blackall et al., 2007; Otero et al., 2018; Riddick et al., 2012). The alteration of soil
biogeochemistry under the disturbance from sea animal activities might have an impact on the
abundance and diversity of the AOA and AOB involved in the nitrogen cycle. Increased bacterial
abundance, diversity, and activity have been detected in penguin or seal colony soils (Ma et al.,
2013; Zhu et al., 2015). Penguin or seal colonies have been confirmed as strong sources for
greenhouse gas $N_2O$ (Zhu et al., 2008, 2013), a by-product of microbial ammonia oxidation





(Santoro et al., 2011). However, the effects of sea animal colonization on AOA and AOB
community structures have not been thoroughly investigated in the maritime Antarctic tundra.

In the present study, we investigated the abundance, activity, and diversity of soil AOA and

AOB in five tundra patches, including a penguin colony, a seal colony, the adjacent animal-lacking
tundra, tundra marsh, and background tundra, where soil biogeochemical properties were
subjected to the differentiating effects of sea animal activities. Our objectives were (a) to examine
the abundance, diversity, and community structure of soil AOA and AOB using the *amoA* gene as
a functional marker; (b) to investigate potential links between *amoA* gene abundance, AOA and
AOB community structures, activity, and environmental variables; and (c) to assess the relative
contribution of these two distinct ammonia-oxidizing groups to nitrification.

## 2 Materials and methods

### 2.1 Study area

The study area is located on the Fildes Peninsula and Ardley Island in the southwest of King

George Island (Fig. 1), having an oceanic climate characteristics. Mean annual air temperature is
about $-2.5\ ^\circ\text{C}$, with a daily mean range from $-26.6$ to $11.7\ ^\circ\text{C}$, and mean annual precipitation is
about 630 mm, mainly in the form of snow. The Fildes Peninsula (about 30 $km^2$ area) is a host to
important sea animal colonies. Based on annual statistical data, the total of over 10,700 sea
animals colonize this peninsula in austral summer. On the western coast are some established seal
colonies including elephant seal (*Mirounga leonine*), weddell seal (*Leptonychotes weddellii*), fur
seal (*Arctocephalus gazella*) and leopard seal (*Hudrurga leptonyx*) (Sun et al., 2004). Ardley
Island, with an area of 2.0 km in length and 1.5 km in width, is connected with the Fildes Peninsula





via a sand dam. This island belong to an important Ecological Reserve for penguin populations in
western Antarctica. A great many of breeding penguins, including Adélie penguins (*Pygoscelis*
*adeliae*), Gentoo penguins (*Pygoscelis papua*), and Chinstrap penguins (*Pygoscelis antarctica*),
colonized on the east of this island in the austral summer. Seal excrements or penguin droppings
rich in nitrogen and phosphorus were transported into local tundra soils by ice-snow melting water
during the breeding period. Mosses and lichens dominate local vegetation. However, the
vegetation is almost absent in penguin or seal colonies because of overmanuring and animal
trampling. More detailed description about the study area can be found in Zhu et al. (2013).
**2.2. Tundra soil collection**
In the summer of 2014/2015, soil samples were collected from the following tundra patches,
as illustrated in Fig. 1:
(i) Penguin colony and penguin-lacking tundra sites: The tundra on Ardley Island was
categorized into three areas from the east to west according to the distance to the penguin nesting
sites (i.e., the intensity of penguin activity): The eastern active penguin colony with nesting sites
PTS (i.e., high penguin-activity area) where penguins have the highest density and high frequency
presence during the breeding period; the adjacent penguin-lacking tundra areas, PLS (i.e., low
penguin-activity areas) in the middle of Ardley Island where penguins occasionally wander and
have a typically low density; and the western tundra marsh, MS, moderately far from penguin
nesting sites (i.e., a slight penguin-activity area) where penguins rarely frequent the sites. In total,
fourteen soil samples were collected from Ardley Island to study the effects of penguin
colonization on the abundance, activity, and community structures of soil AOA and AOB.





Specifically, samples PS1–PS5 were collected sequentially from the center of the colony in the
PTS. Samples PL1–PL4 and MS1–MS5 were randomly collected in the PLS and MS. (ii) The seal
colony and its adjacent tundra sites, STS: These sites are on the western coast of the Fildes
Peninsula. According to the distance to seal wallows (i.e., the intensity of seal activity), samples
SS1–SS5 were collected in sequence to investigate the effects of seal colonization. Site SS1 was
closest to the seal colony (i.e., a high seal-activity site), whereas SS5 was the farthest from the
seal colony (i.e., a low seal-activity site). (iii) Background tundra sites, BS: Three soil samples
were collected from an upland tundra with about 40 m a.s.l. and the distribution of no sea animal
around. The tundra surface is covered with mosses or lichens with a 10–15 cm organic clay layer
(Zhu et al., 2013).
At each sampling site, soil was collected aseptically using a clean scoop from the top 5–10 cm
at the four corners of a 1 m$^2$ subarea, and combined into one sample. Appropriate precautions
were taken to avoid cross-site or human-made contamination. Immediately after collection, each
sample was divided into two portions: one was stored in sterile plastic containers at −80 °C for
the analysis of the microbial community structures, and the other portion was stored at close to
the *in situ* temperature to determine the geochemical characteristics and potential ammonia
oxidation rates. All of the analyses were conducted within one month.
### 2.3. General analysis of soil characteristics
Soil pH was determined by mixing the soil and 1 M KCl solution (1: 3 ratio). Soil moisture
was measured by oven drying at 105 °C to a constant weight. Total nitrogen (TN) and total sulfur
(TS) contents in the soils were determined through a CNS analyzer (vario MACRO, Elementar,



Germany). The chemical volumetric method was used to measure soil total organic carbon (TOC).
The samples were digested in Teflon tubes using $HNO_3$-HCl-HF-$HClO_4$ digestion at 190 °C, and
total phosphorus (TP) was determined using ICP-OES (Perkin Elmer 2100DV, Waltham, MA,
USA). The $NO_3^-$-N, $NO_2^-$-N, and $NH_4^+$-N concentrations were determined through a continuous
flow analyzer (Skalar, Netherlands) (Gao et al., 2018; Zhu et al., 2011).
**2.4. Measurement of soil ammonia oxidation rate**
Potential ammonia oxidation rate (PAOR) in tundra soil was determined using the chlorate
inhibition method (Kurola et al., 2005; Yue, 2007 ). Sodium chlorate was used to inhibit $NO_2^-$
from being oxidized into $NO_3^-$. Briefly, 5 g fresh tundra soil was incubated in 20 ml of 1 mM
phosphate-buffered saline with 1 mM of $(NH_4)_2SO_4$ and $NaClO_3$ in the dark at 15 °C. After
moderately shaking for 24 h, the 5 ml of 2 M KCl was used to extract the nitrite. The optical
density for the supernatant after centrifugation was determined spectrophotometrically at 540 nm.
The standard curve obtained from $NaNO_2$ (0–2.5 µmol $l^{-1}$) was used to calculate the PAOR in the
tundra soils.
**2.5. DNA extraction and gene amplification (PCR)**
Genomic DNA was extracted from 0.25 g of homogenized tundra soils using PowerSoil™
DNA Isolation Kit (Mo Bio, Carlsbad, CA, USA) as described in manufacturer's protocol. The
extracted DNA was eluted in 50 µl of elution buffer, quantified by a Nanodrop-2000
Spectrophotometer (Thermo Scientific, Waltham, MA, USA), and stored at −20 °C. AOA *amoA*
gene fragments (635 bp) were amplified using the primers Arch-amoAF (5'-
STAATGGTCTGGCTTAGACG-3') and Arch-amoAR (5'-GCGGCCATCCATCTGTATGT-3')





(Francis et al., 2005). The *amoA* gene fragment (491 bp) of β-proteobacterial AOB, which
represents known AOB in soil, was amplified using the primer set composed of amoA-1F (5'-
GGGGTTTCTACTGGTGGT-3') and amoA-2R (5'-CCCCTCKGSAAAGCCTTCTTC-3')
(Rotthauwe et al., 1997). All PCR reactions were performed using Taq PCR Master Mix (Sangon
Biotech, Shanghai, China) in a total volume of 50 μl. PCR reactions were carried out with a
thermal profile of 5 min at 95 °C; 35 cycles of 94 °C for 30 s, 56 °C for AOA or 55 °C for AOB
for 45 s, 72 °C for 1 min; and a final 5-min extension cycle at 72 °C (Zheng et al., 2014).
Subsequently, the amplification products were visualized by electrophoresis on 1.0 % agarose gels.
**2.6. Sequencing and phylogenetic analysis**
The amplification products were sent to Sangon Company (Shanghai, China) for purification,
cloning and sequencing (Zheng, 2014) The sequences were edited using DNAstar (DNASTAR,
Madison, WI, USA), and then aligned by muscle using the UPGMB clustering method with the
ClustalX program. The sequences with 97% identity were grouped into one OTU using the
Mothur Program by the furthest neighbor approach (Zheng et al., 2014). The closest reference
sequences were identified at NCBI (http://www.ncbi.nlm.nih.gov/BLAST/) using the BLASTn
tool, and phylogenetic trees were constructed by the neighbor-joining method using the Molecular
Evolutionary Genetics Analysis software (version 5.03). The sequences reported in this study have
been deposited in GenBank under accession unmbers MH318029 to MH318568 and MH301331
to MH302505.



### 2.7. Quantitative real-time PCR


The AOB and AOA *amo*A gene copy numbers for tundra soils were determined in triplicate
using an ABI 7500 Sequence Detection System (Applied Biosystems). The specific details were
given by zheng et al. (2014). The strong linear inverse relationship confirmed the consistency of
the qPCR assay between the threshold cycle and the log value of gene copy numbers ($R^2 = 0.999$
for AOB; $R^2 = 0.997$ for AOA). The amplification efficiencies for AOA and AOB were 99.8 %
and 90.4 %, respectively. Melting curve analysis had only one observable peak at a melting
temperature (Tm) (84.9 °C for AOA and 89.6 °C for AOB) (Supplementary Fig. S3). Negative
controls were subjected to exclude any possible carryover or contamination in all experments.

### 2.8. Statistical analysis


The Shannon–Weiner Index, Simpson Index and the richness estimator Chao 1 were calculated
by the Mothur program (version 1.23.0). The coverage was the percentage of the number of
observed OTUs divided by the Chao 1 (Supplementary Table S2). The Kruskal–Wallis test and
Wilcoxon signed rank test were conducted for the comparison between *amoA* gene abundance and
PAOR from five tundra patches using SPSS Statistics 17 (IBM Corp, Armonk, NY, USA). The
relationships between the ammonia-oxidizer community structure and environmental variables
were explored using canonical correspondence analysis (CCA) in the software Canoco for
windows (version 4.5; Microcomputer Power, Ithaca, NY, USA), because the maximum gradient
length of both AOA and β-AOB was longer than four SD (AOA: 4.406; AOB: 18.326). All
environmental parameter values were transformed into ln(x+1) before statistical analyses. OTU
richness (defined at 3% distance) served as the species input and several simulations of manual





forward selection were performed with 499 Monte Carlo permutations to build the optimal models.
The scaling in the final CCA biplots was focused on inter-sample relations. Correlations between
ammonia-oxidizer gene abundance, diversity, PAOR, and the AOB/AOA ratio with environmental
variables were explored using redundancy analysis (RDA), because the maximum gradient length
was shorter than three SD (AOA: 0.09; AOB: 0.088; PAOR and AOB/AOA: 1.105).

## 3 Results

### 3.1. Soil chemistry and sea animal activities

Overall, almost all the tundra soils were slightly acidic, with a mean pH range of 5.3–6.6.
Penguin and seal colony tundra soils, PTS and STS, had lower TOC contents and C:N ratios than
the animal-lacking tundra soils (PLS), tundra marsh soils (MS), and background tundra soils (BS).
As expected, soil nutrient levels (TN, TP, TS, and $NH_4^+$–N) were higher in PTS, STS, PLS, and
MS than in BS (Table 1). Soil $NH_4^+$–N contents were 1–2 orders of magnitude higher in PTS and
STS than in PLS, MS, and BS, with the means of 176.9 and 137.6 mg $NH_4^+$-N $kg^{-1}$, respectively.
The highest $NO_3^-$-N contents occurred in STS. Phosphorus levels were significantly greater (p <
0.05) in PTS (10.6–32.9 mg $g^{-1}$) than in the other types of tundra soils (mean < 6.0 mg $g^{-1}$). In
the seal colony tundra, soil TOC, TN, TP, TS, and $NH_4^+$-N levels decreased with the distance from
the seal wallow. Likewise, soil TP, TS, and $NH_4^+$-N levels decreased from the eastern penguin
nesting sites to the western tundra marsh. Sea animal activities altered the local soil
biogeochemical properties through the deposition of their excreta, leading to generally low C:N
ratios and a marked increase in soil $NH_4^+$-N and TP contents. Therefore, the soil TP and $NH_4^+$-N



levels and the distance from seal wallows and penguin nesting sites could be used to assess the
intensity of seal or penguin activities.

### 3.2. Gene abundances under sea animal colonization

The abundance of the AOB *amoA* gene was significantly higher (by approximately 2–4 orders
of magnitude) than that of the AOA *amoA* gene (Wilcoxon test, n = 22, P = 0.002) in the penguin
and seal colony and their adjacent tundra soils, PTS, STS, and PLS. However, the abundances of
the *amoA* gene were similar in the MS and BS soils (Fig. 2). Overall, the abundances of AOB and
AOA *amoA* genes were significantly negatively correlated (r = -0.90, P = 0.037) across all the
tundra sites. The archaeal *amoA* gene showed a heterogeneous distribution among the different
tundra patches. AOA *amoA* gene were two orders of magnitude lower in PTS and STS relative to
those in BS and MS. The maximal AOA *amoA* gene abundance appeared in BS, followed by MS
and PLS, whereas the PTS and STS soils had the lowest archaeal *amoA* gene abundances. Soil
AOA *amoA* gene abundances were significantly increased with decreasing animal activity
intensity (i.e., the distance from eastern penguin nesting sites PS1–PS5 to western tundra marsh
MS1–MS5, and from seal wallow site SS1 to the background tundra sites) (Fig. 3).
Unlike the AOA *amoA* genes, AOB *amoA* gene abundances showed the opposite distribution
pattern. The AOB *amoA* gene abundances were significantly higher (by approximately 2–3 orders
of magnitude) in PTS and STS compared with those in MS and BS (Fig. 2). The soil AOB *amoA*
gene abundances increased significantly with increasing animal activities (i.e. the distance from
eastern penguin nesting sites and from the seal wallow) (Fig. 3). The ratios of AOB to AOA *amoA*
copy numbers were strongly affected by animal activities, and were much higher in PTS and STS



than in PLS, MS, and BS (Kruskal–Wallis test, $\chi^2$ = 18.2, P = 0.01). Overall, penguin or seal
activities increases the abundance of soil AOB *amoA* genes, but reduced the abundance of AOA
*amoA* genes, leading to very large ratios ($1.5 \times 10^2$ to $3.2 \times 10^4$) of AOB to AOA *amoA* copy
numbers in PTS and STS. However, the ratios varied only from 0.1 to 7.2 in BS and MS.
**3.3 Potential ammonia oxidation rates under sea animal colonization**
Potential ammonia oxidation rates (PAORs) ranged from 8.9 to 138.8 µg N kg$^{-1}$ h$^{-1}$ in all the
soil samples (Table 1). The PAOR was significantly higher in STS (mean 76.1 µg N kg$^{-1}$ h$^{-1}$) and
PTS (mean 64.7 µg N kg$^{-1}$ h$^{-1}$) than in PLS, MS, and BS (mean 12.0–21.8 µg N kg$^{-1}$ h$^{-1}$; Kruskal–
Wallis test, $\chi^2$ = 11.6, P = 0.02). The PAOR followed the distribution changes of AOB *amoA* gene
abundances, but showed the opposite trend to the AOA *amoA* gene abundances (Fig. 2). A
significant positive correlation ($r^2$ = 0.77, P < 0.001) was observed between the PAOR and the
AOB *amoA* gene abundance when the data from all the tundra patches were combined, whereas
no correlation occurred between PAOR and AOA *amoA* gene abundance (Fig. 4). Therefore, the
AOB populations might contribute more to the PAOR than the AOA populations in the study area.
Interestingly, the PAOR greatly increased with penguin or seal activity intensity, and the greatest
rates occurred at the sites nearest the penguin nests (88.8 ± 2.7 µg N kg$^{-1}$ h$^{-1}$) and seal wallows
(138.8 ± 0.8 µg N kg$^{-1}$ h$^{-1}$) (Fig. 3).
**3.4. Community structure of AOA and AOB under sea animal colonization**
The PCR products were insufficient to construct the clone libraries for the AOA *amoA* gene
from STS and PTS because of the low AOA abundance in the soils, as was the case with the AOB
amoA gene from MS and BS. Overall, 10 AOA and 14 AOB *amoA* gene clone libraries were



successfully constructed. 543 AOA sequences and1175 AOB quality sequences were generated
from the respective sites. Within each individual site, 1–6 AOA OTUs and 6–15 AOB OTUs were
identified, as defined by < 3% divergence in nucleotides. The AOA and AOB OTU numbers for
each library are presented in Table S1.These numbers might be higher if more clones were
sequenced, based on the rarefaction curves (Fig. S1 and Fig. S2). The diversity of the AOB *amoA*
was generally higher than that of AOA *amoA*, based on the indices of Shannon–Wiener and
Simpson. Specifically, the AOA *amoA* gene had higher diversity in PLS and MS than in BS. The
AOB *amoA* gene showed higher diversity in STS and PTS compared with that in adjacent animal-
lacking tundra soils.

The 543 AOA *amoA* gene sequences had 76–100% sequence similarity to each other, and 95–

100% identity with the corresponding top hit *amoA* sequences deposited in GenBank.
Phylogenetic analysis showed that the AOA *amoA* sequences could grouped into 16 unique OTUs,
representing 100% of all the AOA *amoA* OTUs identified, and were affiliated with two
*Nitrososphaera* clusters (Fig. 5a): Cluster I had 11 OTUs and 264 clones, and 57.9% of AOA
*amoA* sequences were from PLS, 41.3% from STS, and only 0.8% from MS. In Cluster II, there
are five unique OTUs and 279 clones, and 58.8% of them were from BS, 38.3% from MS, and
only 2.9% from PLS. Almost all the AOA phylotypes retrieved from PLS and STS were related
to *Nitrososphaera* cluster I, whereas the AOA phylotypes retrieved from MS and BS were
distributed in cluster II (Fig. 6). Seal or penguin activities led to the predominant existence of
AOA phylotypes related to cluster I, but very low relative abundances in AOA phylotypes related
to cluster II, which were almost completely excluded in STS and PLS. Almost all AOA phylotypes





in BS and MS were related to *Nitrososphaera* cluster II, whereas the relative abundances of AOA
phylotypes related to cluster I were very low or undetectable.

The 1175 AOB *amoA* gene sequences shared 87–100% sequence identity to each other, and

93–100% identity with the closest matched GenBank sequences. Phylogenetic analysis showed
that the AOB *amoA* sequences could be grouped into 38 unique OTUs, representing 58.5% of all
the AOB *amoA* OTUs identified, and these *amoA* sequences were grouped into four clusters
according to the evolutionary distance of the phylogenetic tree with known sequences from AOBs
in the *Nitrosospira* genera (Fig. 5b). Cluster I had 11 OTUs and 226 clones, and 67.7% of AOB
*amoA* sequences were from PTS, 23.5 % from STS, 8.4% from PLS, and only 0.4% from MS.
There are 17 unique OTUs and 521 clones in clusters II and III. The sources of the OTUs in cluster
II were similar to those of cluster I, with 69.8% from PTS, 29.9% from STS, and 0.3% from PLS.
For cluster III, 79.2% of the sequences were from PLS, 19.8% from STS, and 1.0% from MS.
Cluster IV had nine unique OTUs and 370 clones from PLS (50.0%), STS (36.8%) and MS
(13.2%), respectively. Of all the AOB phylotypes retrieved from PTS were related to dominant
*Nitrosospira* clusters I and II, whereas AOB phylotypes related to cluster III and IV were
completely excluded because of strong penguin activity (Fig. 6). The AOB phylotypes retrieved
from STS were distributed in clusters I, II, III, and IV (16–38% for each cluster). Almost all the
AOB phylotypes retrieved from PLS and MS were related to *Nitrosospira* clusters III and IV.
**3.5. Relationships of the ammonia-oxidizer community structure with environmental variables**

The relationships of the AOA and AOB communities with environmental variables were

analyzed using CCA. The environmental variables explained 58.4% of the total variance in the





AOA *amoA* genotype compositions, and 66.8% of the cumulative variance of the genotype-
environment relationships in the first two CCA dimensions (Fig. 7a). Overall, the AOA
community structures significantly correlated with C:N, TOC, and $NO_3^-$-N in tundra soils (Table
2), and the combination of the three factors explained 60.3% of the variation. Although other
environmental parameters, including TP, pH, and $NH_4^+$-N were not statistically significant ($P >$
0.05), these variables additionally explained 26.5% of the variation. The AOA richness and
phylotypes were evidently inhibited in STS and PLS because seal or penguin activities. However,
high soil C:N and TOC concentrations increased the AOA richness and phylotypes in MS and BS.
As illustrated in Fig. 7b, the first two dimensions explained 26.6% of the total variance in the
AOB compositions, and 54.3% of the cumulative variance of the AOB genotype-environment
relationships. The composition and distribution of AOB communities correlated significantly with
$NH_4^+$-N and C:N ratios, and the two factors combined yielded 21.9% of total CCA explanatory
power. The others including TP, $NO_3^-$-N and pH accounted for 27.1% of the variance. Penguin or
seal activities significantly increased the AOB richness and phylotypes in STS and PTS through
higher $NH_4^+$-N and P input from sea animal excrement, whereas AOB richness and phylotypes
were closely related to the soil C:N in PLS and MS.
Correlations among *amoA* gene abundance, diversity, PAOR, and the ratios of AOB:AOA
abundance with environmental variables were examined via Redundancy Analysis (RDA) (Fig.
8). The AOA *amoA* gene abundance and diversity were positively related to the C:N ratio (P =
0.002), and negatively correlated with $NH_4^+$-N (P = 0.004). Two factors combine yielded 63.5%
of the total RDA explanatory power (Table S2). Higher soil C:N increased the AOA abundance
and diversity in BS and MS, but higher $NH_4^+$-N input inhibited their abundance and diversity in



PLS and STS because of penguin or seal activities. Significant correlations were obtained between
AOB *amoA* gene abundance, diversity, and environmental factors including the C:N ratio (P =
0.004), TOC (P = 0.012), and $NH_4^+$-N (P = 0.05). These three factors combined yielded 73.2% of
the total explanatory power (Table S3). The ratios of AOB to AOA and PAOR showed positive
correlations with $NH_4^+$-N (P = 0.002), TP (P = 0.046), and TS (P = 0.030), but negative
correlations with the C:N ratio (P = 0.002) and TOC (P = 0.048). These factors explained 87.5%
of the variation (Table S4). Compared with those in BS and MS, penguin or seal activities
significantly increased the AOB *amoA* gene abundance, diversity, PAOR, and the ratios of AOB
to AOA in STS, PTS, and PLS because of the increase in $NH_4^+$-N and TP input from animal
excrement.

### 339    4 Discussion

### 340    4.1. Effects of sea animal colonization on AOA and AOB abundances

In this study, soil AOA *amoA* gene abundances were two orders of magnitude lower in PTS
and STS relative to BS and MS; however, AOB *amoA* gene abundances were approximately 2–3
orders of magnitude higher in PTS and STS than in MS and BS, indicating that sea animal
activities increased the AOB population size, but inhibited AOA abundances in tundra soils (Fig.
2 and Fig. 3). Overall, the archeal *amoA* gene abundances obtained here were similar to the
abundance range reprted in the soils of the Antarctic Dry Valleys and arctic tundra soils; however,
the bacterial *amoA* gene abundances were two to three orders of magnitude higher in PTS and
STS than in Antarctic Dry Valleys (Alves et al., 2013; Magalhães et al., 2014). In contrast to
previous studies indicating that AOA were more abundant than AOB in some terrestrial or marine



ecosystems (Beman et al., 2008; Lam et al., 2007; Wuchter et al., 2006; Yao et al., 2011), and in
soils from Antarctic Peninsula (Jung et al., 2011), our qPCR estimates showed that the bacterial
*amoA* copy numbers were much greater than those of archeal *amoA* in PTS, STS and PLS because
of sea animal activities. However, their abundances were very close to each other in BS and MS.
The ratios of AOB to AOA abundance were strongly affected by sea animal activities. A shift in
the relative abundance of AOA and AOB recorded previously for the Antarctic Dry Valleys, with
a greater abundance of AOB compared with that of AOA for Battleship Promontory and Miers
Valley, and the reverse for Upper Wright Valley and Beacon Valley (Magalhães et al., 2014). The
results for PTS, STS, and PLS are also in agreement with those detected in subglacial soils (Boyd
et al., 2011).

The ratios of AOB to AOA showed significant positive correlations with $NH_4^+$-N, TP, and TS

when all the data were combined in the five tundra patches (Fig. 8). This suggested that $NH_4^+$-N,
TP, and TS are key factors when bacterial *amoA* genes are much more abundant than archeal *amoA*
genes. In Antarctica, the productivity of terrestrial ecosystems is strongly limited because of the
extremely low nitrogen levels (Park et al., 2007). However, the physiochemical properties for
tundra soils were strongly influenced by the deposition of penguin or seal excreta under effects of
local microbes (Tatur et al., 1997). Sea animals provide considerable external N inputs for their
colony soils and adjacent tundra soils through direct input of their excreta and atmospheric
deposition via ammonia volatilization (Lindeboom, 1984; Sun et al., 2002; Blackall et al., 2007;
Zhu et al., 2011; Riddick et al., 2012). Like ammonium, P and S are typical elements in penguin
guano, and they have been used to indicate penguin activity intensity (Sun et al., 2000).
Significantly elevated $NH_4^+$–N and TP concentrations occurred in PTS and PLS compared with





those in BS (Table 1). These conditions may be beneficial for nitrification, allowing high
abundance and diversity of bacterial *amoA*, which explains the strong correlations between AOB
abundances and $NH_4^+$–N, TP, and TS in the sea animal colony soils (Fig. 8). This is agreed with
the high bacterial diversity and abundance previously documented in penguin or seal colony soils
and ornithogenic sediments (Ma et al., 2013; Zhu et al., 2015).
The AOA abundance and diversity showed a positive correlation with C:N in tundra patches,
but a significant negative correlation with $NH_4^+$-N levels (Fig. 8). AOA might better adapt to low
$NH_4^+$ and oligotrophic environments because the half-saturation constant for ammonia oxidation
by *Thaumarchaeota* is lower than that by AOB (Martens-Habbena et al., 2009). High $NH_4^+$-N
concentrations might partially inhibit AOA populations (Hatzenpichler et al., 2008). This result is
similar to that reported for some agricultural soils with increased fertilization, and grassland soils
with increased grazing (Fan et al., 2011; Prosser and Nicol, 2012; Pan et al., 2018), supporting the
conclusion that AOA and AOB generally inhabit different niches in soil, distinguished by the
$NH_4^+$ concentration and availability (Verhamme et al., 2011; Wessén et al., 2011).

### 4.2. Effects of sea animal colonization on soil ammonia oxidation rates

In this study, PAOR ranged from 9 to 139 µg N $kg^{-1}$ $h^{-1}$, which was lower than nitrification
rates measured in most agricultural soils (83–1875 µg N $Kg^{-1}$ $h^{-1}$) (Fan et al., 2011; Ouyang et
al., 2016; Daebeler et al., 2017). One reason might be the selection of a 15 °C incubation
temperature, which is lower than the incubation temperatures used in other studies. Generally, the
gross nitrification rate and *amoA* abundance increased significantly when the incubation
temperature was higher than 15 °C (Daebeler et al., 2017; Zhao et al., 2014). Notably, comammox



*Nitrospira* may actually compete with ammonia oxidizers for ammonium, after which comammox
oxidize ammonia to nitrate on their own via a one-step process (Daims et al., 2015; van Kessel et
al., 2015). In this study, the method of measuring nitrification rates did not include the activity of
these organisms because sodium chlorate was used to prevent $NO_2^-$ from being oxidized to $NO_3^-$,
whereas other methods likely capture the comammox activity (Santoro, 2016). Our measurements
indicated that there were significant differences in the PAOR across different tundra patches (P =
0.02), and the PAORs in STS and PTS were about 10 times higher than those in BS and MS. A
significant correlation was observed between the PAOR and $NH_4^+$–N, TP, and sulfur (Fig. 8).
Overall, ammonia oxidation activity was modulated by soil biogeochemical processes under the
disturbance of sea animal activities: sufficient input of the nutrients $NH_4^+$–N, TP, and TS from sea
animal excreta.
The gene abundance of AOB *amoA* was markedly higher that of AOA *amoA*, and AOA found
it difficult to tolerate the high ammonium environment in PTS, STS, and PLS, indicating that
AOB might play a more important role in nitrification. In agreement with these results, AOB
dominated nitrification in the areas where it was easy to achieve nitrogen input, whereas the
relative contribution of AOA to nitrification was higher in the areas where the ammonium
concentration remained low (Fan et al., 2011; Sterngren et al., 2015). Moreover, the cell-specific
activity for AOB was 10 times higher than that for AOA due to the bigger cell size of AOB
(Hatzenpichler et al., 2012; Prosser and Nicol, 2012). Therefore, AOB might play a more
important role in nitrification in STS, PTS, and PLS compared with that in BS and MS.





In addition, AOA might play a role that cannot be ignored in MS and BS, just like the
prevalence of AOA among ammonia-oxidizers in Arctic soils (Alves et al., 2013; Daebeler et al.,
2017). AOB groups were mostly undetectable in the analysis of MS and BS. Although unknown
γ-AOB groups might not have been detected, the primer set used here covers the β-AOB groups
typically found in soils (Alves et al., 2013). The BS and MS were covered with lush tundra plants
and were rich in organic carbon (Table 1), which has been shown to favor AOA because their
substrates can be provided through the mineralization of soil organic matter (Stopnišeket al., 2010;
Habteselassie et al., 2013).
**4.3. Effects of sea animal colonization on genotypic diversity of soil AOA and AOB**
In this study, distinct AOA communities appear to inhabit different types of tundra patches,
depending on sea animal activities (Fig. 5). It was difficult to amplify the AOA *amoA* gene from
STS and PTS, whereas a high diversity of AOA *amoA* genes was observed in PLS, MS and BS.
Phylogenetic analysis indicated that the AOA *amoA* sequences in Cluster I were from PLS and
tundra soils close to seal wallows, while the sequences in Cluster II were from BS and MS (Fig.
6). AOA in most extreme environments have lower levels of microbial diversity than benign
ecosystems because of the requirement for specific physiological adaptations, which allow
organisms to exploit the combination of physical and biochemical stressors (Cowan et al., 2015).
Cluster I found in the PLS might represent AOA adapting to survival in the presence of relatively
high soil nutrients, for which the presence of the *amoA* gene represents either secondary
metabolism or an ancestral remnant no longer active because of the high AOB abundance in these
areas. Detected OTUs in Cluster I had their closest matches mainly from the hyper-arid soils of



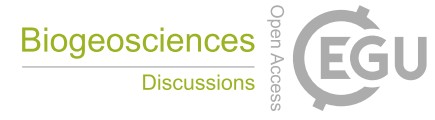

Antarctic dry valleys (Magalhães et al., 2014), wetland soils (Zheng et al., 2014), alpine meadow
soils (Zhao et al., 2017), and some agricultural soils (Glaser et al., 2010). Cluster II were more
prevalent in BS and MS, probably because of their stronger adaptation to barren soil environments.
In cluster II, the sequences were affiliated with sequences recovered from cold environments,
including the soils of Tibetan Plateau (Zhang et al., 2017) and Icelandic grassland soils (Lam et al.,
2009). The compositions of soil AOA populations are likely not to be explained by single
physicochemical properties, and their community structures significantly correlated with tundra
soil C:N, TOC, and $NO_3^-$-N, which was consistent with previous studies (Glaser et al., 2010;
Wessén et al., 2011).
The AOB *amoA* gene generally had a higher diversity than AOA, similar to results in the
Antarctic Dry Valley soils (Magalhães et al., 2014). A high diversity of AOB *amoA* gene occurred
in STS, PTS and PLS compared to BS, indicating that penguin or seal activities had important
effects on AOB genotypic diversity. According to the evolutionary distance of the phylogenetic
tree, AOB *amoA* sequences were grouped into four clusters with known sequences from the
*Nitrosospira* genera, and they are in the lineages of *Nitrosospira sp.* En13 (EF175097),
*Nitrosospira sp.* LT1FMf (AY189144), *Nitrosospira sp.* EnI299 (EF175100), *Nitrosospira sp.* III7
(AY123829), and *Nitrosospira sp.* Wyke8 (EF175099) (Fig. 5b). The sequences in clusters I and
II were mainly from PTS and STS, and the detected OTUs in Cluster I had their closest matches
from mixed community culture systems, meadow to forest transect in Oregon Cascade Mountains
(Mintie et al., 2003), and Dutch agricultural soils and reservoir sediments (Silva et al., 2012). For
Clusters III and IV, the sequences were predominantly from PLS and STS, and they were affiliated
with sequences recovered from high altitude wetland (Shan et al., 2014). Previous studies have





shown that multiple environmental factors affected the AOB communities (Dang et al., 2008;
Mosier and Francis, 2008). In this study, the $NH_4^+$-N concentrations seemed to be the most
important factors influencing the AOB community structure, which was in accordance with the
results from different environments (Bouskill et al., 2012; Jung et al., 2011; Li et al., 2015).
Moreover, the C:N ratio and TP also affected the AOB *amoA* community compositions (Zheng et
al., 2013). Therefore, the AOB community compositions were impacted by the biogeochemical
factors related to sea animal activities, such as sufficient supply of the nutrients $NH_4^+$–N and TP
from sea animal excreta.

## 5 Conclusions

The findings of this study concerning the abundance, activity, and diversity of tundra soil AOA
and AOB provide insights into microbial mechanisms driving nitrification in maritime Antarctica.
We confirmed the presence of AOA and AOB *amoA* genes in five different tundra patches, and
demonstrated that the spatial distribution heterogeneities of the tundra soil AOA and AOB
communities were driven by penguin or seal activities. The soil AOB *amoA* copy numbers were
generally higher than the AOA *amoA* copy numbers, following the higher PAOR in penguin or
seal colonies and their adjacent tundra, compared with that in the background tundra and marsh
tundra, which are moderately far away from the animal colonies. Penguin or seal activities resulted
in significant shift of soil AOA and AOB community compositions. The diversity of the AOB
*amoA* gene was greater in STS and PTS than in PLS and MS, and the majority of the AOB
sequences were closely related to *Nitrosospira*-like sequences. The archaeal *amoA* gene had
higher diversity in STS, PLS, and MS than in BS, and they were associated with sequences





recovered from barren soils. Soil AOB and AOA abundances, and their community compositions,
were related to soil biogeochemical processes under the disturbance of sea animal activities, such
as soil C:N alteration, and a sufficient supply of the nutrients $NH_4^+$–N, N and P from animal
excreta. Overall, this study significantly enhanced the understanding of ammonia-oxidizing
microbial communities in tundra environment of maritime Antarctica.

## Acknowledgments

This work was supported by the NSFC (Grant No. 41576181; 41776190). We are particularly
grateful to the members of Chinese Antarctic Research Expedition and Polar Office of National
Ocean Bureau of China for their support and timely help. The final derived data presented in this
study are available at https://doi.org/10.5281/zenodo.1260292.

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



**Table 1.** Soil properties, potential ammonia oxidation rates, ammonia oxidizer populations, and diversity for the soil samples (n = 28) that span a penguin colony, a seal colony, and their adjacent animal-lacking tundra across Ardley Island and the Fildes Peninsula in maritime Antarctica.

| Sampling No. | pH | Moisture (%) | TOC (%) | Nitrogen (%) | Sulfur (%) | TP (mg/g) | NH4+-N (mg/Kg) | NO3-N (mg/Kg) | NO2-N (mg/Kg) | PAOR (µgN Kg-1 h-1) | AOA (copies g-1) | AOB (copies g-1) |
|---|---|---|---|---|---|---|---|---|---|---|---|---|
| Seal colony tundra soils (STS) | | | | | | | | | | | | |
| SS1 | 4.8 | 0.31 | 4.88 | 1.21 | 0.34 | 3.6 | 650.9 | 4.6 | 0.1 | 138.8±0.8 | $1.79\times10^5$ | $9.22\times10^8$ |
| SS2 | 8.2 | 0.33 | 6.62 | 1.69 | 0.48 | 5.0 | 17.7 | 19.1 | 0.7 | 115.3±15.5 | $3.99\times10^4$ | $5.92\times10^5$ |
| SS3 | 4.6 | 0.20 | 0.39 | 0.09 | ND | 1.3 | 17.9 | 61.7 | 0.2 | 8.9±0.5 | -- | $3.85\times10^8$ |
| SS4 | 5.2 | 0.18 | 0.65 | 0.13 | 0.08 | 1.2 | 0.6 | 12.1 | ND | 38.4±5.1 | $5.53\times10^4$ | $2.57\times10^8$ |
| SS5 | 5.4 | 0.27 | 1.16 | 0.13 | 0.07 | 0.8 | 1.1 | 13.9 | ND | 79.3±44.5 | -- | $3.03\times10^7$ |
| Mean±SE | 5.6±0.6[ab] | 0.26±0.03[ab] | 2.74±1.13[a] | 0.65±0.30[ab] | 0.24±0.08[ab] | 2.4±0.7[a] | 137.6±114.8[a] | 22.3±9.1[a] | 0.3±0.12[a] | 76.1±21.4[a] | $(9.1\pm2.7)\times10^{4a}$ | $(4.0\pm1.4)\times10^{8ab}$ |
| Active penguin colony tundra soils (PTS) along the coast on Ardley Island | | | | | | | | | | | | |
| PS1 | 5.7 | 0.65 | 8.91 | 1.45 | 0.44 | 10.6 | 151.4 | 2.5 | 0.3 | 88.8±2.7 | $5.95\times10^4$ | $7.54\times10^8$ |
| PS2 | 5.9 | 0.53 | 4.39 | 0.80 | 0.16 | 12.5 | 461.0 | 1.7 | 0.6 | 70.9±14.4 | $2.49\times10^4$ | $4.62\times10^8$ |
| PS3 | 4.9 | 0.27 | 10.24 | 1.55 | 0.41 | 23.7 | 59.9 | 7.2 | 0.2 | 48.9±0.4 | $1.28\times10^4$ | $4.13\times10^8$ |
| PS4 | 5.2 | 0.66 | 12.90 | 1.79 | 0.31 | 32.9 | 21.4 | 4.3 | 0.7 | 41.1±2.7 | $2.44\times10^4$ | $3.21\times10^8$ |
| PS5 | 4.9 | 0.25 | 4.92 | 0.83 | 0.38 | 18.1 | 190.7 | 54.7 | 0.9 | 17.3±2.1 | $1.57\times10^4$ | $4.25\times10^8$ |
| Mean±SE | 5.3±0.2[a] | 0.47±0.08[b] | 8.27±1.44[abc] | 1.28±0.18[ab] | 0.34±0.04[b] | 19.6±3.6[b] | 176.9±69.1[a] | 14.1±9.1[a] | 0.5±0.12[a] | 53.4±11.0[ac] | $(2.7\pm0.7)\times10^{4a}$ | $(4.8\pm0.7)\times10^{8a}$ |
| The middle penguin-lacking tundra soils (PLS) on Ardley Island | | | | | | | | | | | | |
| PL1 | 6.7 | 0.86 | 12.10 | 1.15 | 0.26 | 5.7 | 3.7 | 1.3 | ND | 19.8±1.2 | $2.58\times10^5$ | $7.94\times10^7$ |
| PL2 | 6.6 | 0.42 | 4.12 | 0.39 | 0.07 | 8.1 | 5.7 | 1.2 | ND | 16.2±0.5 | $4.69\times10^5$ | $2.09\times10^7$ |
| PL3 | 6.6 | 0.95 | 28.59 | 2.53 | 0.31 | 3.1 | 3.4 | 13.2 | ND | 33.1±0.9 | $1.75\times10^4$ | $5.03\times10^7$ |
| PL4 | 6.5 | 0.85 | 6.52 | 0.72 | 0.18 | 5.4 | 1.2 | 2.5 | ND | 18.3±1.4 | $1.40\times10^5$ | $1.24\times10^8$ |
| Mean±SE | 6.6±0.1[b] | 0.77±0.10[c] | 12.83±4.77[abc] | 1.20±0.41[ab] | 0.21±0.05[ab] | 5.6±0.9[a] | 3.5±0.8[b] | 4.5±2.5[a] | - | 21.8±3.3[bc] | $(5.4\pm2.6)\times10^{5b}$ | $(6.9\pm0.2)\times10^{7b}$ |



**The western tundra marsh soils (MS) on Ardley Island**

| | | | | | | | | | | | | |
|---|---|---|---|---|---|---|---|---|---|---|---|---|
| MS1 | 6.1 | 0.66 | 8.64 | 0.89 | 0.25 | 5.2 | 1.1 | 10.3 | 0.1 | 15.5±1.2 | $3.46\times10^6$ | $3.11\times10^5$ |
| MS2 | 5.7 | 0.84 | 20.35 | 1.59 | 0.20 | 1.8 | 1.2 | 7.8 | 0.4 | 8.9±2.2 | $2.39\times10^6$ | $1.73\times10^7$ |
| MS3 | 5.1 | 0.86 | 20.88 | 1.98 | 0.26 | 1.8 | 11.5 | 9.8 | 0.4 | 10.3±1.5 | $1.33\times10^5$ | $9.97\times10^4$ |
| MS4 | 5 | 0.92 | 32.32 | 2.66 | 0.24 | 2.2 | 11.5 | 13.1 | 0.3 | 14.4±3.9 | -- | $4.93\times10^4$ |
| MS5 | 5.1 | 0.93 | 25.45 | 2.35 | 0.25 | 1.9 | 5.3 | 12.0 | 0.3 | 10.8±3.4 | $3.80\times10^5$ | $2.44\times10^5$ |
| Mean±SE | 5.4±0.2[ab] | 0.84±0.04[c] | 21.53±3.46[c] | 1.89±0.28[b] | 0.24±0.01[ab] | 2.6±0.6[a] | 6.1±2.1[b] | 10.6±0.8[a] | 0.3±0.1[a] | 12.0±1.1[b] | $(2.1\pm0.6)\times10^6$[b] | $(5.9\pm3.5)\times10^6$[c] |

**Background tundra soils (BS) on the upland of the Fildes Peninsula**

| | | | | | | | | | | | | |
|---|---|---|---|---|---|---|---|---|---|---|---|---|
| BS1 | 5.3 | 0.16 | 16.82 | 0.48 | 0.12 | 2.4 | 1.1 | 23.6 | 0.5 | 12.8±1.5 | $4.33\times10^6$ | $2.16\times10^7$ |
| BS2 | 5.6 | 0.18 | 18.12 | 0.51 | 0.08 | 1.9 | 0.7 | 16.4 | 0.5 | 17.6±0.5 | $7.94\times10^6$ | $2.39\times10^6$ |
| BS3 | 5.3 | 0.20 | 17.55 | 0.43 | 0.05 | 3.0 | 1.2 | 16.4 | 0.6 | 11.1±0.8 | $1.56\times10^7$ | $1.11\times10^7$ |
| Mean±SE | 5.4±0.1[ab] | 0.35±0.01[a] | 17.50±0.31[b] | 0.47±0.02[a] | 0.08±0.02[a] | 2.5±0.3[a] | 2.3±0.1[b] | 16.7±2.0[a] | 0.5±0.1[a] | 13.8±1.6[bc] | $(9.3\pm2.7)\times10^6$[b] | $(1.2\pm0.5)\times10^7$[c] |



**Table 2.** Individual and combined contributions of soil biogeochemical properties to the AOA and AOB community structures in tundra patches.

|  | Soil properties | F | P | Individual contribution |
|---|---|---|---|---|
| AOA | C/N | 2.815 | 0.014 | 17.7% |
|  | TOC | 2.337 | 0.018 | 9.7% |
|  | $NO_3^-$ | 2.165 | 0.034 | 8.3% |
|  | $NH_4^+$ | 0.983 | 0.466 | 9.3% |
|  | TP | 1.012 | 0.442 | 4.6% |
|  | pH | 1.653 | 0.094 | 4.5% |
|  | Combined effect of all factors |  |  | 87.4% |
| AOB | C/N | 1.844 | 0.002 | 6.1% |
|  | $NH_4^+$ | 1.823 | 0.002 | 6.9% |
|  | TP | 1.39 | 0.078 | 11.6% |
|  | pH | 1.383 | 0.066 | 9.1% |
|  | $NO_3^-$ | 1.161 | 0.258 | 10.7% |
|  | Combined effect of all factors |  |  | 48.9% |




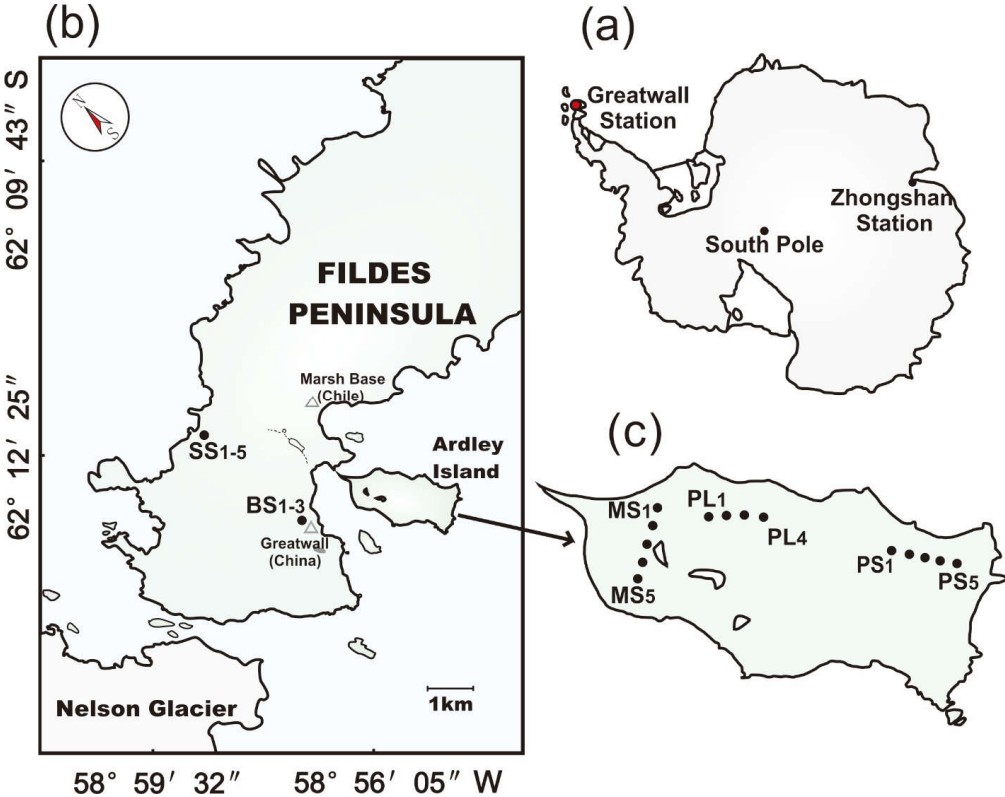

**Figure 1**. Study area and soil sampling sites. Panel (a), the red dot indicates the location of the investigation area in maritime Antarctica. Panel (b), location of the sampling sites on the Fildes Peninsula. The sampling soils from tundra patches included the active seal colony tundra soils STS (SS1–5) in the western coast of the Fildes Peninsula, and the background tundra soils on the upland areas (BS1–3). Panel (c), the location of the sampling sites on Ardley Island. The sampling soils from tundra patches included the western tundra marsh soils (MS1–5), the eastern active penguin colony tundra soils PTS (PS1–5) and the adjacent penguin-lacking tundra soils PLS (PL1–4). Note: The map was drawn using CorelDRAW X7 software (http://www.corel.com/cn/).



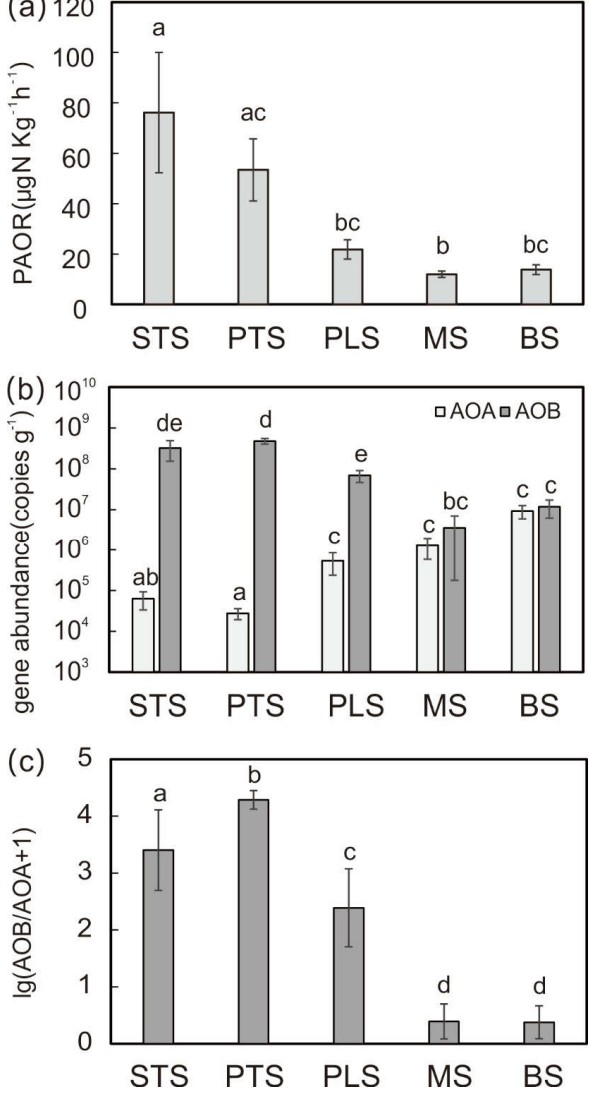

**Figure 2**. Comparisons of soil potential ammonia oxidation rates (PAOR) (a); AOA and AOB

*amoA* gene copy numbers (b); and log ratio of AOB: AOA abundances (c); between five tundra

patches. The error bars indicate standard deviations of the means.



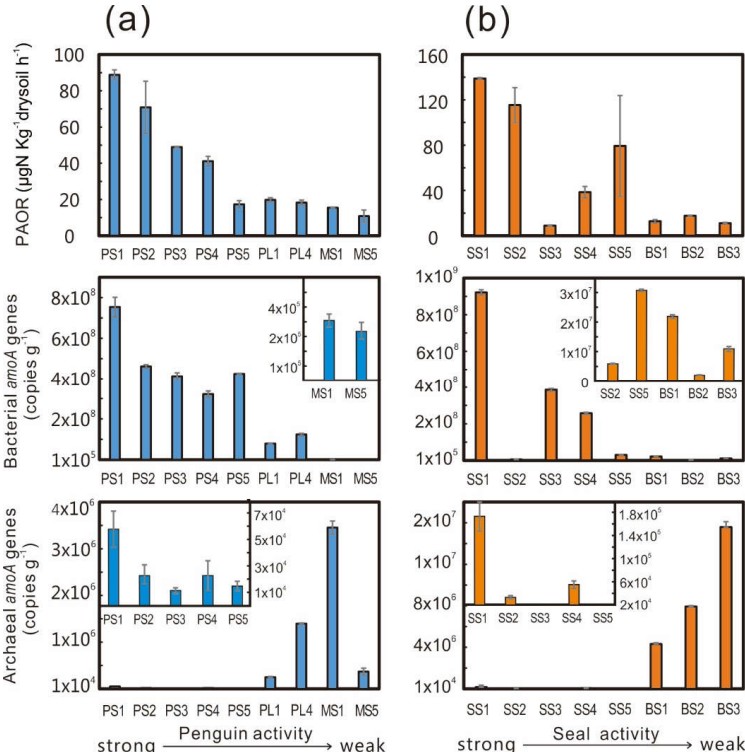

**Figure 3**. Effects of penguin or seal activity on potential ammonia oxidation rates (PAORs), and

AOA and AOB *amoA* gene copy numbers in tundra soils. (a) Penguin colonies and their

adjacent tundra; (b) Seal colonies and their adjacent tundra. The error bars of potential ammonia

oxidation rates indicate the standard deviations of triplicate incubations, whereas the error bars

of the *amoA* copy numbers indicate standard deviations of triplicate real-time PCR assays.





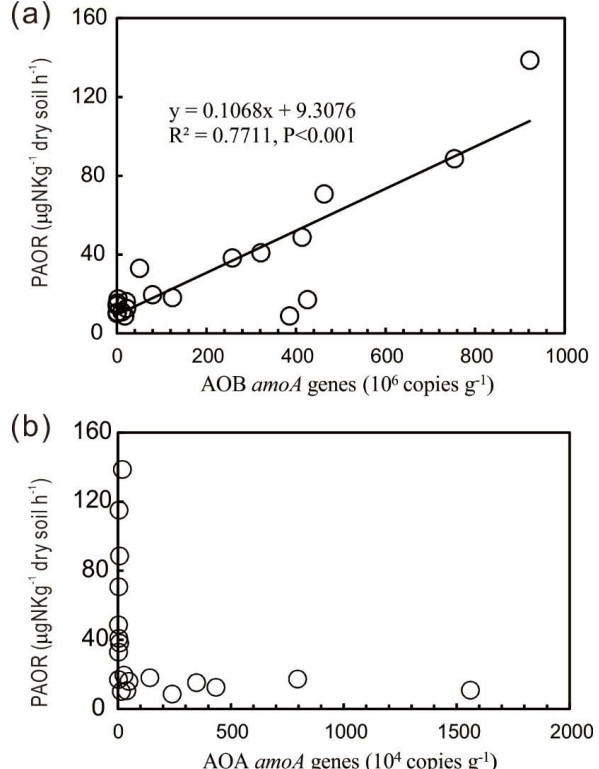

**Figure 4**. Correlation between potential ammonia oxidation rates (POARs) and AOA and AOB

*amoA* gene copy numbers in tundra soils of maritime Antarctica.



**Figure 5**. Neighbor-joining phylogenetic tree of AOA *amoA* (a) and AOB *amoA* (b). The phylogeny is based on nucleotide sequences. Bootstrap values ≥ 50% (of 1000 iterations) are shown near the nodes. GenBank accession numbers are shown for sequences from other studies. OTUs were defined at 97% similarity. Numbers in parentheses following each OTU indicate the number of sequences recovered from each sampling site.

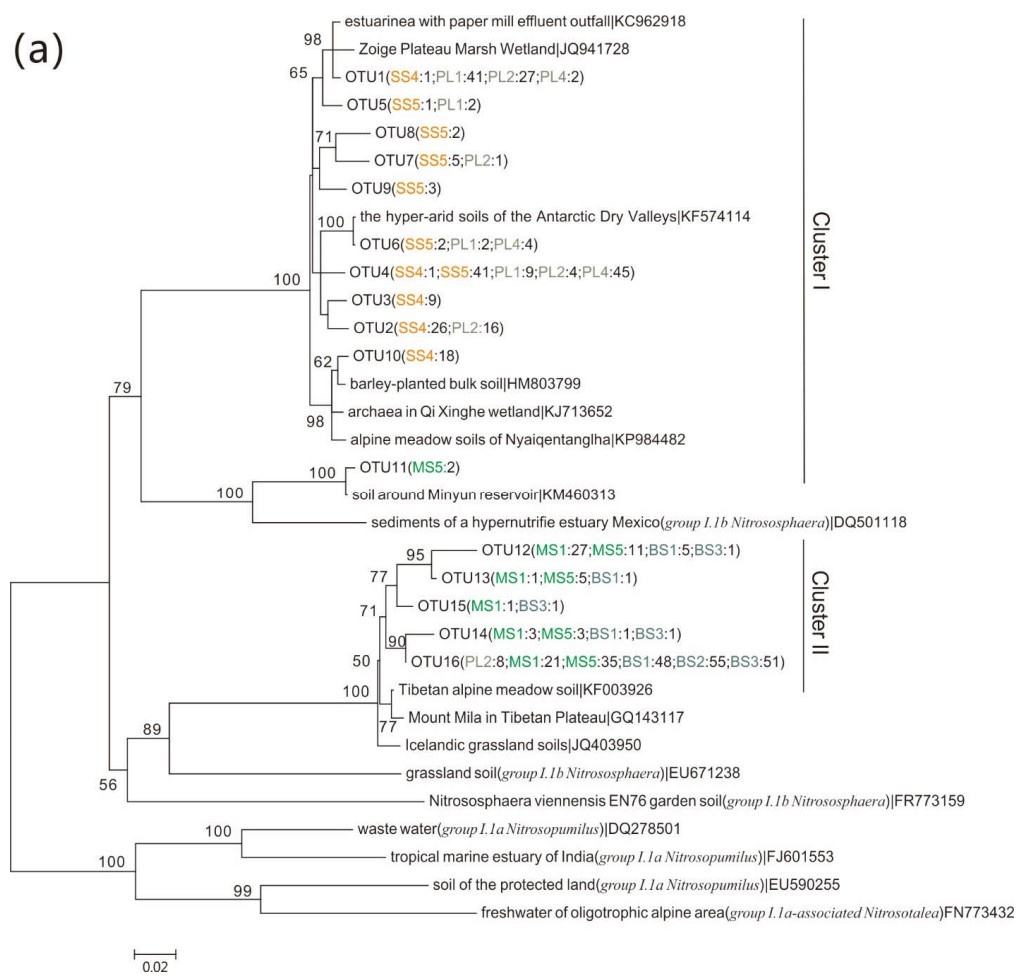



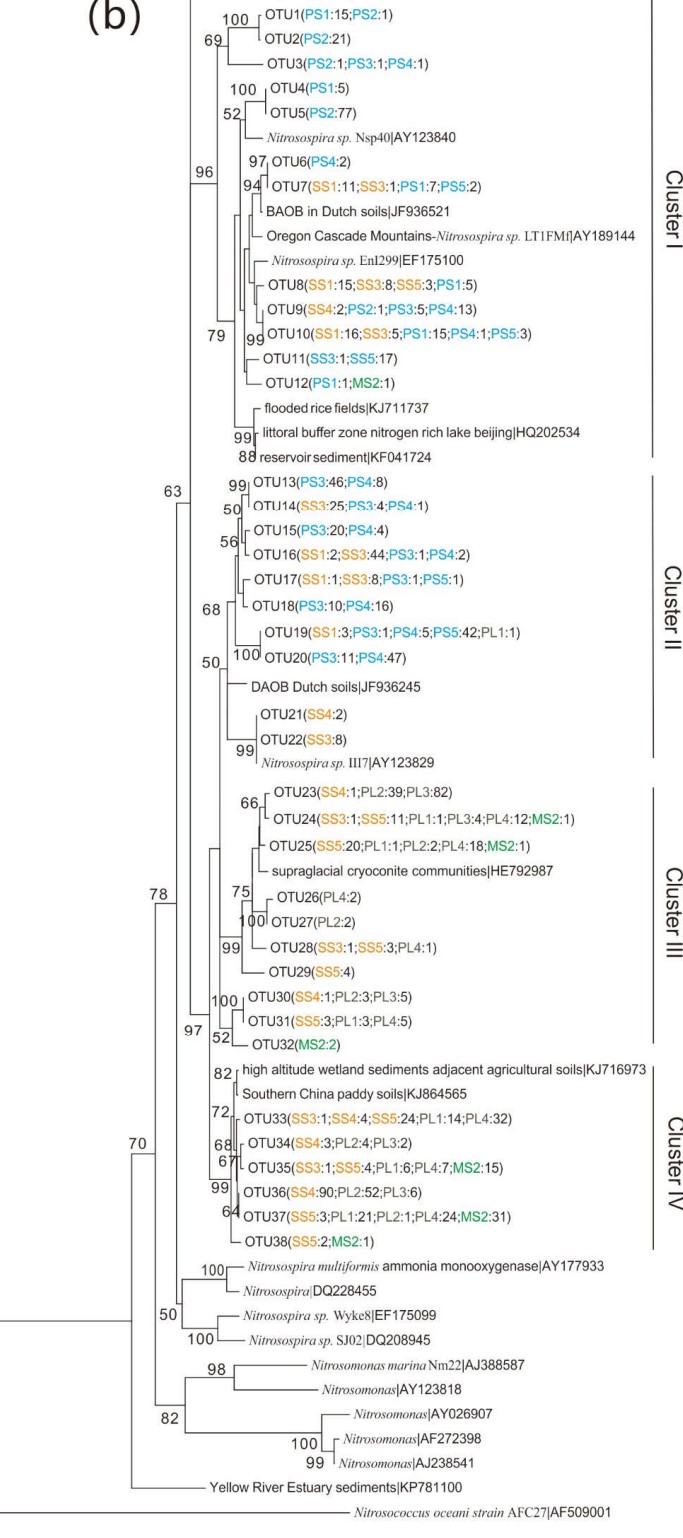



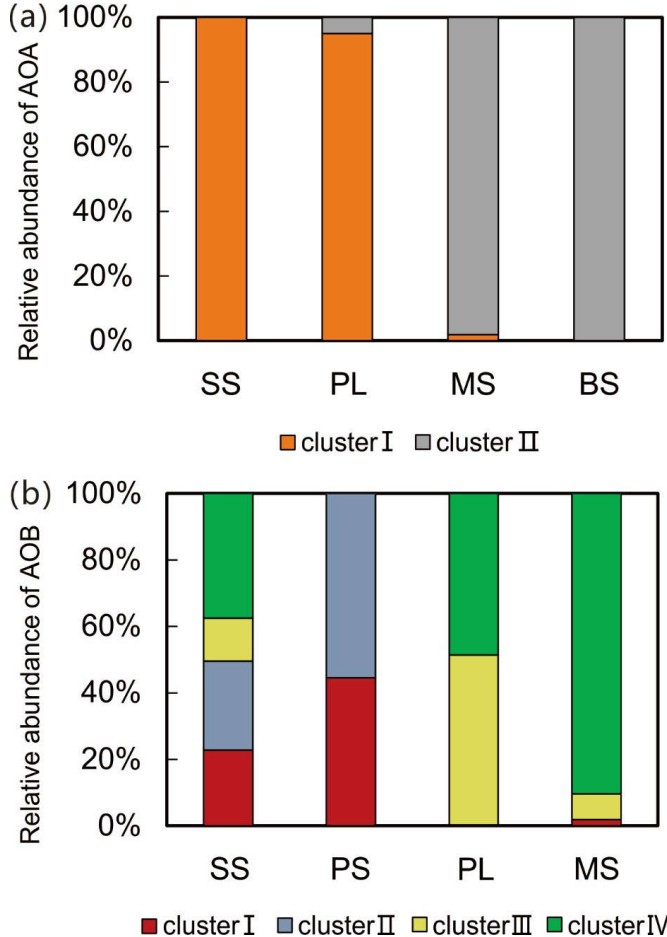

**Figure 6**. Relative abundance of partial AOA (a) and AOB (b) sequences retrieved from five

tundra patch soils subjected to different effects of sea animal activities, as related to different

*Nitrososphaera* or *Nitrosospira* clusters.

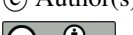



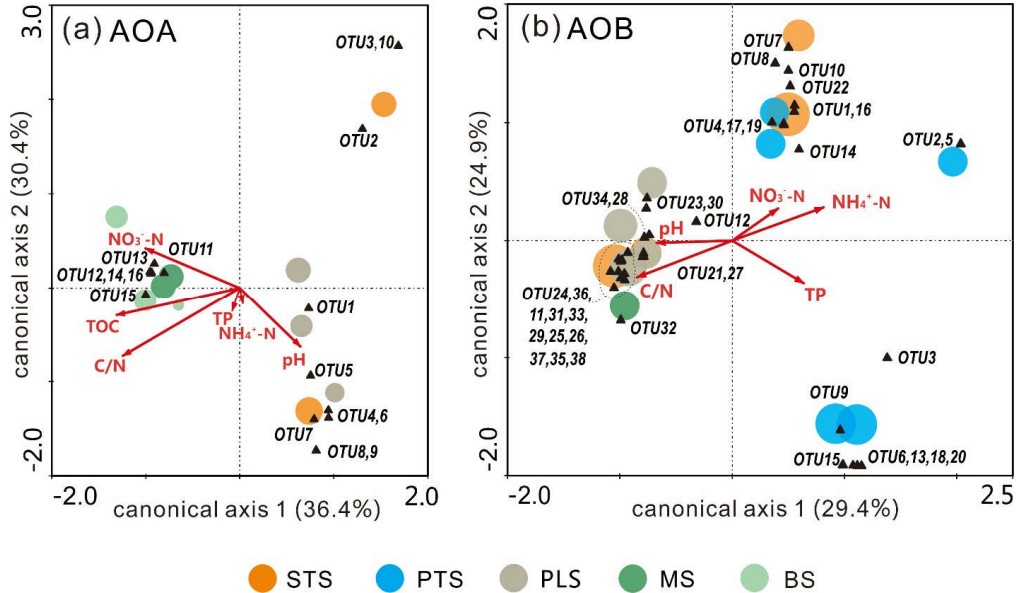

**Figure 7.** Canonical correspondence analysis (CCA) ordination plots for the relationship between the AOA and AOB community structures with environmental variables. The circles with different colors represent the various sampling sites. The size of the circles corresponds to the OTU richness in individual samples. The black triangles represent amoA phylotypes. Environmental variables are represented by red arrows. The percentage of species-environment relation variance explained by the two principal canonical axes is represented close to the axes.



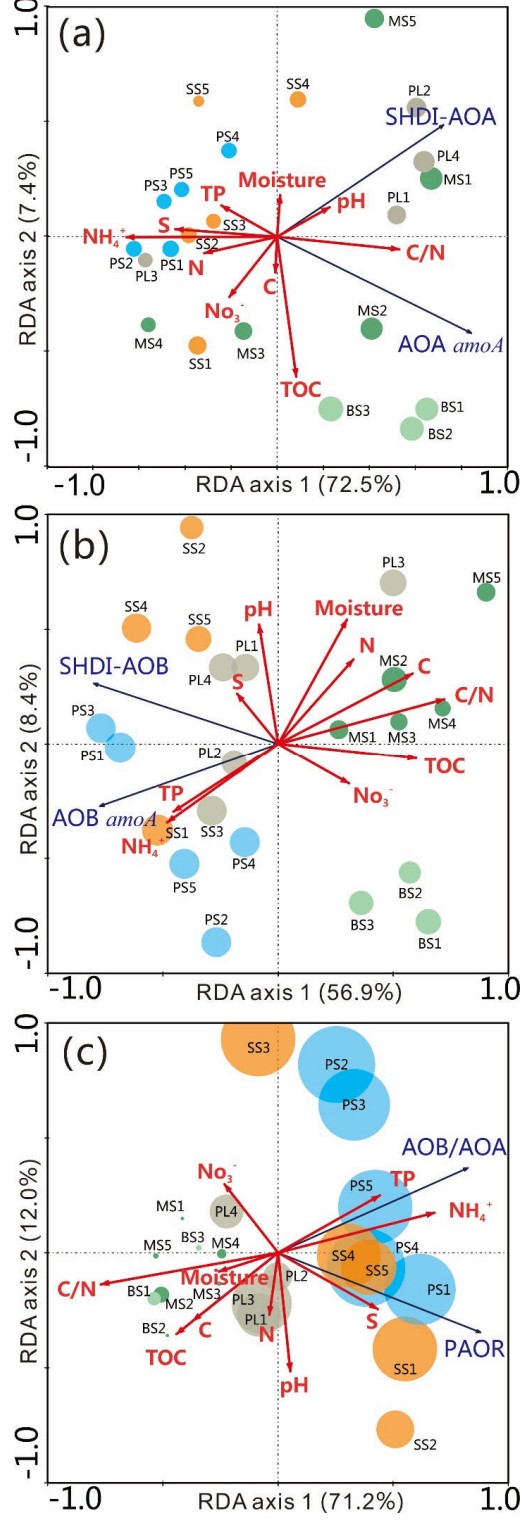



**Figure 8.** Redundancy analysis ordination (RDA) plots for the relationships between copy number and Shannon's diversity index of AOA *amoA* genes and environmental variables (a), between copy number and Shannon's diversity index of AOB *amoA* genes and environmental variables (b), and between the PAOR, ratio of AOB/AOA *amoA*, and environmental variables (c). Gene copy log values for AOA, AOB, and log ratios of AOB/AOA *amoA* are represented as circles whose diameters are scaled linearly to the magnitude of the value. In the RDA ordination diagram, the angle and length of the arrow relative to a given axis reveals the extent of the correlation between the variables and the canonical axis (environmental gradient). SHDI indicates Shannon's diversity index.