# Peer review of "Effects of Sea Animal Colonization on the Coupling between Dynamics and"

_Biogeosciences, 2019_

## Referee Comment (RC1) · Anonymous Referee #1 · 19 Apr 2019

The manuscript "Effects of Sea Animal Colonization on the Coupling between Dynamics and Activity of Soil Ammonia-oxidizing Bacteria and Archaea in Maritime Antarctica" investigated the abundance, diversity, community structure and bioactivity of ammonia oxidising bacteria and archaea in Antarctic maritime soils. colonized by sea animals. The results found that soils colonized by seals and peguines exhibited higher AOB than AOA, as well as higher ammonia oxidizing rates than the control tundra soils. These findings suggest that AOB may play a more important role than AOA in driving ammonia oxidizing in peguine and seal colonized soils, while AOA more important in

control tundra soils. The research provided very interesting findings, which contributes to understand the nitrogen cycling in Antarctic coastal soils. Here are my concerns: 1. Too many abbreviations for samples and sites, authors got PS, PL, MS, SS and BS for sample IDs and PTS, PLS, MS, STS and BS for sites. This is just too confusing to read. 2. Line 25, Nitrosospira is an AOB, Nitrososphaera is an AOA, need to change their order in the sentence. 3. Line 41, Comammox (COMplete AMMonia OXidiser) is an abbreviation, please provide its full name. 4. line 40-41, "Only recently. . .", this sentence seems to be out of picture, I would suggest to remove it. 5. Line 56, "However, there has been limited. . .", I don't think this sentence is correct, especially after the authors listed so many studies on ammonia-oxidisers in line 52-56. 6. Line 210ïïjŇ "mean pH range of 5.3-6.6", The word mean and range seems contradict to each other, I guess the work "mean" here represents the mean of each sampling site. This is better to be clarified. 7. Line 211, "Penguin and seal colony tundra soils, PTS and STS, had lower TOC. . ." Firstly I couldn't find the C:N ratio in table 1; secondly the table 1 used id SS, PS, PL etc, but main text used PTS, STS etc, therefore lacking consistency, lastly, the TOC level of PS (PTS) site was not significantly different from the PLS, MS and BS sites. I think the lack of significance was due to large variations? 8. Line 213, "as expected, soil nutrient levels. . .", why is this expected? I could understand that TN may be higher with penguin guano and seal faeces input, but why TP and TS? Furthermore, there was no significant TN difference in BS with SS, PS and PL, similarly for TS, TP, and even ammonia. This greatly reduces the reliability of authors' claim. After a close inspection on the numbers provided in the table 1, it seems that the large ammonia in SS and PS was due to a single sample in each site, I don't know how far SS1 and SS2 are to generate such large differences. Furthermore, this may not make much sense, the SS1 has ammonia concentration of 650 mg/kg, the highest among all other SS samples and 35 times higher than SS2, but its total nitrogen was only 1.2%, even 0.4% lower than SS2. Similar unusual pattern was also in the ammonia concentration in PS2 sample I would strongly suggest the authors to recheck their measurements. As these environmental factors are the basis of many statistical analysis performed

later, this would completely make authors conclusion invalid. 9. Line 219, "likewise, soil . . .", which site is author referring here? PTS or PLS? Or stating a generally pattern from PTS, PLS to MS? Please clarify. As PTS is clearly not showing this pattern.. 10. Line 222, "therefore, the soil TP and NH4. . ." this is a very bold statement, and lacking proof. Something like linear regression would be required. 11. Line 229, (fig. 2), figure 2 has 3 parts (a, b and c), please specify which part of the figure 2 is referred to. 12. Line 229, "overall. . ." please provide a scatter plot to visualise this (can be put in supplementary) 13. Line 231, " the archaeal amoA gene showed a heterogeneous distribution" what does heterogeneous distribution mean? 14. Line 232, there was a mixed usage of AOA amoA and archaeal amoA in the manuscript, please make them consistent. 15. Line 237, "fig 3", similar to a previous comment, there are 6 parts of figure 3, please specify which part does this refer to. 16. Line 235, "Soil AOA amoA gene abundances were significant. . ." This statement is inappropriate, I would agree that animal activity reduces archaeal amoA gene abundance, but the statement of increasing archaeal gene abundance with reduced animal activity need a better proof. A correlation analysis between the activity intensity index and archaeal amoA gene abundance would be required. 17. Line 240, "The soil AOB amoA gene abundances increased . . ." this is incorrect, author stated that the order of sampling reflected the intensity of seal activity (highest in SS1 and lowest in SS5) (line 123-127), but clearly the abundance of bacterial AOB gene reduced with reduced penguin or seal activity. 18. line 242 "The ratios of AOB to AOA amoA. . ." please cite figure 2c for this sentence. 19. line 250, " The PAOR was significantly higher in STS. . .", this is not fully correct, the PAOR of PS samples was not significantly different from the BS site. 20. Figure 3, again, archaeal results appeared before bacterial results, thus their figure should appear before bacterial figures. 21. Line 258, "Interestingly, the PAOR. . ." Please confirm this statement with a statistical analysis, as PAOR increased from SS3 to SS5. 22. Line 271, "Specifically, the AOA amoA gene . . ." please present these results as a table or a figure. 23. Line 276 "Phylogenetic analysis showed that the AOA. . ." Why and how phylogenetic analysis was used to group sequences into OTUs? In addition, the entire

sentence is confusing, please revise. 24. Line 289, "Phylogenetic analysis showed that AOB amoA..."Why and how phylogenetic analysis was used to group sequences into OTUs? In addition, the entire sentence is confusing, please revise. 25. Line 312, "The AOA richness and phylotypes were evidently inhibited..." what does this mean? The richness of AOA is indeed lower in STS and PLS, but this result has already been presented in line 269. 26. Line 323 why RDA was used to investigate the correlation among amoA gene abundance, diversity and etc? I would think RDA is used to deal with matrix dataset, but all these variables are vector variable. If only correlations were required, Pearson or partial Pearson correlation would be sufficient. If the contribution of each variable is required, I would think VPA analysis would be a better option. 27. Line 325, "The AOA amoA gene abundance...", which type of correlation is this? Please report the r value, and may be also scatter plots in the supplementary. Furthermore, authors stated that both AOA amoA gene abundance and diversity were related to C:N ratio, but only one P-value was reported. 28. Table 1 need to provide full name of site, also the site codes do not match those in the main text. 29. Figure 2. The order of figure need to change, Figure 2b appeared first in the manuscript, and they should appear first in figure 2.

––––––––––––––––––––––––––––––––

---

## Referee Comment (RC2) · Anonymous Referee #2 · 7 Jun 2019

The manuscript entitled "Effects of sea animal colonization on the coupling between dynamics and activity of soil ammonia-oxidizing bacteria and archaea in maritime Antarctica" by Wang et al. describes the effect of sea animal colonization on the community composition of ammonia oxidizers. The subject matter is interesting and the work in general is technically sound, however, my main concern is that the authors make claims about the relationship between nitrification rates and ammonia-oxidizer dynamics. Furthermore, there are some inconsistencies within the environmental parameter data, as well as very speculative parts in the discussion which need to be addressed.

[Figure]

General comments: The authors measured potential ammonia oxidation rates by adding 1mM NH4Cl and incubating the samples at 15 degrees, which seems to be very artificial and far from in situ rates. It is highly speculative to comment on in-situ ammonia oxidation rates based in these measurements. Hence, assessing the relative contribution of AOA and AOB to nitrification rates based on the presented measurements is highly speculative and can only be suggested based on the differences in abundance between those two groups. Further, the authors talk about "inhibition" of AOA due to seal and penguin activities (e.g., lines 312-313, line 344), however, the presented data simply suggests a higher abundance of AOB over AOA. While the environmental conditions might be more favorable for AOB, it is highly speculative to assume that this is caused by inhibition and should be phrased more carefully.

The ammonia concentrations of the 5 samples within the same site are sometimes extremely variable (e.g. 650 vs 0.1 in the STS site). How far were the different sampling points apart? Some of the data in Table 1 seems surprising or/and might be not well represented, e.g. the sum of the percentage of total carbon, nitrogen and sulfur makes up e.g. only 0.5%. What are the other 99.5%? Reporting total carbon, nitrogen and sulfur in mg/kg might be more useful as well. Additionally, the abbreviations of the sites are not very intuitive and easy to confuse.

Specific comments: Line 25: Nitrosospira are AOB and Nitrososphaera are AOA, needs to be switched. Line 32-33: "The results provide insights into the mechanism how microbes drive nitrification in maritime Antarctica", here again the authors make claims that are not supported by the presented data. The mechanisms of nitrification are not studied. Line 37: "biogeochemical nitrogen cycle" instead of "biogeochemical cycle for nitrogen" Line 40: AOB were discovered much earlier than 2015, please chose a different reference Line 41: comammox should be spelled out Line 46: Are you referring to the marine water column or sediments? Please specify (instead of mentioning "marine layers") and add the appropriate references. Line 93: "daily mean range" is contradictory, please correct. Line 101: "A great many" should probably read

"A great majority" Line 346: typo in "reported" Lines 378-380: This statement is not necessarily correct. There might be more diversity within Km's of AOA that differ from that of N. maritimus. Making such a claim based on a single organism is very speculative. Lines 393-397: The connection with comammox is not very intuitive. Did you detect comammox? Also, the reference of Santoro 2016 does not fit here because it measures actual rates (instead of potential rates) using stable isotopes in marine environments where no comammox has been found thus far. Lines 417-420: Why does a high organic carbon favor AOA over AOB? So far most studies have shown that AOA are inhibited by complex organic substrates (Stieglmeier et al 2011, Qin et al 2017, etc). Lines 430-433: This statement is highly speculative and likely wrong. Why would the presence of an amoA gene be an ancestral remnant that is not active? There is no data presented supporting such claims. Lines 446-455: this section does not discuss the data and should be moved to results

---

## Author Comment (AC2) · 4 Jul 2019

Please read the supplement for our revised manuscript and Supplimentary Material. The manuscript is uploaded in the form of a pdf document.

Please also note the supplement to this comment:
https://www.biogeosciences-discuss.net/bg-2019-114/bg-2019-114-AC2-supplement.pdf

---

## Author Response (AR1)

**Response to Associate Editor**

*Dear Dr. Denise M. Akob,*

We are very grateful to you for your valuable comments about the revision of this manuscript (MS No.bg-2019-114). We have revised the manuscript carefully according to your comments, and provided a version of track changes in the re-submitted files.

**The detailed responses inserted into associate editor's comments are attached as follows:**
*1. In the response to reviewers you provide a table that compares your soil physiocochemical parameters (e.g., TOC, TN) to previous work by your group. The earlier studies have different samples names. Can you make the naming consistent over the different papers so that the readers can compare?*

**Author response:** In the response to reviewers, we re-measured soil TC and TN concentrations based on the same soil samples collected in 2015, and provided a table that compared the measurement results, which are similar to those in this paper. In our previous papers, the soil samples were collected by different people in different years, thus it is very difficult to make the naming consistent over the different papers although these soils were collected in the same study area. However, **the readers can still clearly compare the concentration differences of soil physiochemical parameters in different tundra areas according to our previous published papers and this paper.**

*2. Define OTU at first usage*

**Author response:** OUT has been defined as "operational taxonomic unit" on **line 182** in the revised manuscript with track changes.

*3. mothur is actually lower case and this reference to the program needs to be included: Schloss PD, Westcott S, Ryabin T, Hall J, Hartmann M, Hollister E, Lesniewski R, Oakley B, Parks D, Robinson C, Sahl J, Stres B, Thallinger G, Van Horn D, Weber C. 2009. Introducing mothur: Open-source, platform-independent, community-supported software for describing and comparing microbial communities. Appl Environ Microbiol 75(23):7537-7541, doi: http://dx.doi.org/10.1128/AEM. 01541-09.*

**Author response:** We have revised the error of initial, and added the reference. Please see **line 183, 201** in the revised manuscript.

*4. Please include the reference for BLAST (Madden T. 2002. The BLAST Sequence Analysis Tool. In McEntyre J, Ostell J (ed), The NCBI Handbook. National Center for Biotechnology Information, Bethesda, Maryland USA.)*

**Author response:** The reference for BLAST has been added. Please see **line 184** in revised manuscript.

*5. Is Figure 6 really needed for the main paper? I feel like it could move to supplemental information. Up to you. Or is there a way to combine the information in Figure 5 and 6?*

**Author response:** Considering that some of the information provided in Figure 6 has been shown in Figure 5, therefore Figure 6 has been moved to Supplementary Material in the revised manuscript.

*6. Other minor revisions*

**Author response:** The incorrect words and phrases have been corrected in the revised manuscript.

<h1 style="text-align:center">Response to Referee #1</h1>

At first, we would like to express our appreciations to you for your kind help and valuable comments about the revision of this manuscript (MS No.bg-2019-114). We have considered your valuable suggestions and carefully revised this manuscript.

**The detailed responses inserted into reviewer #1 comments are attached as follows:**

*Anonymous Referee #1*

*The manuscript "Effects of Sea Animal Colonization on the Coupling between Dynamics and Activity of Soil Ammonia-oxidizing Bacteria and Archaea in Maritime Antarctica" investigated the abundance, diversity, community structure and bioactivity of ammonia oxidising bacteria and archaea in Antarctic maritime soils colonized by sea animals. The results found that soils colonized by seals and peguines exhibited higher AOB than AOA, as well as higher ammonia oxidizing rates than the control tundra soils. These findings suggest that AOB may play a more important role than AOA in driving ammonia oxidizing in penguine and seal colonized soils, while AOA more important in control tundra soils. The research provided very interesting findings, which contributes to understand the nitrogen cycling in Antarctic coastal soils. Here are my concerns:*

**Author response:** Thanks for your positive comments and valuable suggestions.

*1. Too many abbreviations for samples and sites, authors got PS, PL, MS, SS and BS for sample IDs and PTS, PLS, MS, STS and BS for sites. This is just too confusing to read.*

**Author response:** Thanks for your good suggestions. In the revised manuscript, we have used SS, PS, PL, MS, and BS for the samples and sites consistently to escape the ambiguity.

*2. Line 25, Nitrosospira is an AOB, Nitrososphaera is an AOA, need to change their order in the sentence.*

**Author response:** The order has been changed in the sentence. Please see **line 26-27** in revised manuscript with track changes.

*3. Line 41, Comammox (COMplete AMMonia OXidiser) is an abbreviation, please provide its full name. 4. line 40-41, "Only recently…", this sentence seems to be out of*

picture, *I would suggest to remove it*.

**Author response:** Thanks for your good suggestions. This sentence and comammox has been removed in the revised manuscript.

*5. Line 56, "However, there has been limited…", I don't think this sentence is correct, especially after the authors listed so many studies on ammonia-oxidisers in line 52-56.*

**Author response:** This sentences has been removed in the revised manuscript.

*6. Line 210 "mean pH range of 5.3-6.6", The word mean and range seems contradict to each other, I guess the word "mean" here represents the mean of each sampling site. This is better to be clarified.*

**Author response:** This has been corrected into "Almost all the tundra soils were slightly acidic, and the mean pH ranged from 5.3 to 6.6 at each tundra patch". Please see **line 220-221** in revised manuscript.

*7. Line 211, "Penguin and seal colony tundra soils, PTS and STS, had lower TOC…" Firstly I couldn't find the C:N ratio in Table 1;*

**Author response:** The data about C:N ratios have been added in **Table 1**.

*secondly the table 1 used id SS, PS, PL etc, but main text used PTS, STS etc, therefore lacking consistency,*

**Author response:** In the revised manuscript, we have used id SS, PS, PL, MS and BS in **both Table 1 and the main text** for their consistency.

*lastly, the TOC level of PS (PTS) site was not significantly different from the PLS, MS and BS sites. I think the lack of significance was due to large variations?*

**Author response:** Yes. The lack of significance might be due to large variations of TC contents caused by high soil heterogeneity in each tundra patch. Generally, penguin or seal colonies and the active areas are devoid of vegetation due to toxic overmanuring and their trampling. Penguin and seal colony tundra soils, PS and SS, had lower TC contents and C:N ratios than the animal-lacking tundra soils (PL), tundra marsh soils (MS), and background tundra soils (BS).

*8. Line 213, "as expected, soil nutrient levels…", why is this expected? I could understand that TN may be higher with penguin guano and seal faeces input, but why*

*TP and TS? Furthermore, there was no significant TN difference in BS with SS, PS and PL, similarly for TS, TP, and even ammonia. This greatly reduces the reliability of authors' claim.*

**Author response: (1)** According to food chains, krill, as main food for penguins, is rich in N, P and S, whereas penguin is one of main foods for seals. The N, P, and S are highly enriched in penguin guano, and they are typical elements for penguin guano (**Sun et al., 2000; Sun et al., 2004; Zhu et al., 2013; Zhu et al., 2014**). Therefore soil nutrients N, P and S are higher in penguin or seal colony soils due to the deposition of penguin guano or seal excrements in maritime Antarctica; **(2)** Generally, penguin or seal colonies and the active areas are devoid of vegetation due to toxic overmanuring and trampling (**Tatur et al., 1997; Sun et al., 2004**), whereas the animal-lacking tundra areas adjacent to penguin or seal colonies, with moderate amount of nutrients, is favorable for vegetation, such as mosses and algae, due to the volatilization and deposition of ammonia and sulfur-containing compounds from penguin guano or seal excreta. **The growth and nitrogen fixation of the vegetation, and the volatilization and deposition of ammonia and sulfur-containing compounds** increased soil TC, TN and TS contents in animal-lacking tundra soils (**Zhu et al., 2011; Zhu et al., 2013**). In penguin or seal colonies, the penguin or seal populations showed high inhomogeneous distribution, and this led to the large differences in soil TC, TN, TS, $NH_4^+$-N contents. **Therefore, overall mean TC, TN and TS contents showed no significant differences between SS, PS, PL and BS (Table 1).** We have revised the corresponding part in the manuscript, please see the **line 221-229**.

The related references are as follows:

Tatur, A., Myrcha, A., and Niegodzisz, J.: Formation of abandoned penguin rookery ecosystems in the maritime Antarctic, Polar Biology, 17, 405–417, https://doi.org/10.1007/s003000050135, 1997.

Sun, L. G., Xie, Z. Q., and Zhao, J. L.: Palaeoecology: A 3,000-year record of penguin populations, Nature, 407, 858, https://doi.org/10.1038/35038163, 2000.

Sun, L. G., Liu, X. D., Yin, X. B., Zhu, R. B., Xie, Z. Q., and Wang, Y. H.: A 1,500-year record of Antarctic seal populations in response to climate change, Polar Biology, 27, 495–501, https://doi.org/10.1007/s00300-004-0608-2, 2004.

Zhu, R. B., Liu, Y. S., Xu, H., Ma, D. W., and Jiang, S.: Marine animals significantly increase tundra $N_2O$ and $CH_4$ emissions in maritime Antarctica, Journal of Geophysical Research: Biogeosciences, 118(4), 1773–1792, https://doi.org/10.1002/2013JG002398, 2013.

Zhu, R. B., Sun, J. J., Liu, Y. S., Gong, Z. J., and Sun, L. G.: Potential ammonia emissions from penguin guano, ornithogenic soils and seal colony soils in coastal Antarctica: effects of freezing-thawing cycles and selected environmental variables, Antarctic Science, 23(1), 78–92, https://doi.org/10.1017/S0954102010000623, 2011.

Zhu, R. B., Wang, Q., Ding, W., Wang, C., Hou, L. J., and Ma, D. W.: Penguins significantly increased phosphine formation and phosphorus contribution in maritime Antarctic soils, Scientific Reports, 4, 7055, https://doi.org/10.1038/srep07055, 2014.

*After a close inspection on the numbers provided in the table 1, it seems that the large ammonia in SS and PS was due to a single sample in each site, I don't know how far SS1 and SS2 are to generate such large differences. Furthermore, this may not make much sense, the SS1 has ammonia concentration of 650 mg/kg, the highest among all other SS samples and 35 times higher than SS2, but its total nitrogen was only 1.2%, even 0.4% lower than SS2. Similar unusual pattern was also in the ammonia concentration in PS2 sample I would strongly suggest the authors to recheck their measurements. As these environmental factors are the basis of many statistical analysis performed later, this would completely make authors conclusion invalid.*

**Author response:** We have rechecked the measurement results, and confirm that our data are right and valid. The reasons are as follows:

**(1)** We measured soil TC and TN concentrations again, **which are provided in the following Table**, and the results are similar to those in this study, and their concentrations still showed large differences at the each sites within penguin or seal colony; **(2)** We have measured soil physiochemical properties several times which were given in our previous published papers:

Zhu RB, Liu YS, Xu H, Ma DW, Jiang S. Marine animals significantly increase tundra $N_2O$ and $CH_4$ emissions in maritime Antarctica. Journal of Geophysical Research: Biogeosciences, 2013, 118: 1773–1792, doi:10.1002/2013JG002398.

Zhu RB, Liu YS, Ma ED, Sun JJ, Xu H, Sun LG. Nutrient compositions and potential greenhouse gas production in penguin guano, ornithogenic soils and seal colony soils in coastal Antarctica. Antarctic Science, 2009, doi:10.1017/S0954102009990204.

Soil chemical properties, especially $NH_4^+$-N, $NO_3^-$-N, P and S concentrations, also showed large differences due to effects of penguin or seal activities according to the two papers above.

Therefore we think that TC, TN, TS, TP, $NH_4^+$-N and $NO_3^-$-N levels showed high heterogeneity in penguin or seal colony tundra soils, PS and SS **due to the deposition**

of penguin or seal excreta, and the differences of tundra vegetation and soil texture caused by animal tramp.

| | Original No. | No. in the paper | Detection in 2015 | | | Re-detection in 2019 | | |
|---|---|---|---|---|---|---|---|---|
| | | | N(mg/g) | C(mg/g) | C/N | N(mg/g) | C(mg/g) | C/N |
| Seal colony soils in western coast on Fildes Peninsula | SK1 | SS1 | 12.12 | 48.67 | 4.02 | 9.99 | 54.52 | 5.51 |
| | SK4 | SS2 | 16.94 | 70.06 | 4.13 | 13.38 | 81.81 | 6.15 |
| | SK6 | SS3 | 0.87 | 5.56 | 6.37 | 1.51 | 10.34 | 6.85 |
| | SK7 | | 2.40 | 13.64 | 5.69 | The sample is used up. | | |
| | SK8 | SS4 | 1.28 | 8.59 | 6.71 | The sample is used up. | | |
| | SK9 | | 2.63 | 18.88 | 7.19 | 2.51 | 18.98 | 7.56 |
| | SK10 | SS5 | 1.30 | 11.54 | 8.87 | The sample is used up, the same as below | | |
| Penguin colony soils on Ardley Island | E1 | | 10.54 | 50.58 | 4.8 | 8.68 | 55.83 | 6.43 |
| | E2 | PS1 | 14.55 | 84.65 | 5.82 | | | |
| | E3 | | 7.73 | 51.64 | 6.68 | 7.92 | 55.84 | 7.05 |
| | E4 | PS2 | 8.34 | 38.08 | 4.56 | | | |
| | E5 | | 15.07 | 89.71 | 5.95 | 13.48 | 92.33 | 6.85 |
| | E6 | PS3 | 17.90 | 120.76 | 6.75 | | | |
| | E7 | | 27.33 | 156.78 | 5.74 | 26.34 | 162.93 | 6.19 |
| | E8 | PS4 | 15.45 | 107.47 | 6.96 | | | |
| | E9 | | 9.99 | 73.10 | 7.31 | 8.87 | 79.72 | 8.99 |
| | E10 | PS5 | 7.97 | 45.82 | **5.75** | | | |
| The middle tundra soils on Ardley Island | M1 | PL1 | 11.53 | 117.64 | 10.2 | 9.88 | 124.91 | 12.64 |
| | M2 | | 13.61 | 138.41 | 10.17 | | | |
| | M3 | PL2 | 3.93 | 38.05 | 9.68 | 4.51 | 50.41 | 11.18 |
| | M4 | | 8.09 | 82.40 | 10.18 | | | |
| | M5 | PL3 | 25.30 | 302.52 | 11.96 | 23.94 | 301.93 | 12.61 |
| | M6 | | 20.19 | 222.45 | 11.02 | | | |
| | M7 | PL4 | 7.17 | 71.85 | 10.02 | 6.37 | 74.82 | 11.75 |
| | M8 | | 9.84 | 114.99 | 11.69 | | | |
| | M9 | | 11.47 | 110.65 | 9.65 | | | |
| | M10 | | 15.84 | 177.48 | 11.21 | 15.69 | 190.83 | 12.16 |
| | M11 | | 11.61 | 119.29 | 10.27 | | | |
| | M12 | | 4.34 | 44.40 | 10.23 | | | |
| | M13 | | 9.65 | 116.36 | 12.05 | | | |

| Group | ID | Sub | | | | | | |
|---|---|---|---|---|---|---|---|---|
| | M14 | | 3.33 | 30.13 | 9.04 | 2.77 | 30.49 | 11.01 |
| | M15 | | 12.95 | 147.59 | 11.39 | | | |
| The tundra marsh soils in west of Ardley Island (almost no animals) | W1 | MS1 | 8.93 | 95.54 | 10.7 | 9.65 | 111.82 | 11.59 |
| | W2 | | 11.92 | 148.81 | 12.49 | | | |
| | W3 | MS2 | 15.89 | 193.95 | 12.2 | 14.35 | 191.57 | 13.35 |
| | W4 | | 17.83 | 217.76 | 12.21 | | | |
| | W5 | | 12.93 | 141.64 | 10.95 | 10.79 | 136.73 | 12.67 |
| | W6 | MS3 | 19.79 | 226.90 | 11.46 | | | |
| | W7 | | 10.81 | 122.84 | 11.37 | 9.37 | 122.43 | 13.07 |
| | W8 | MS4 | 26.57 | 355.02 | 13.36 | | | |
| | W9 | | 21.88 | 254.01 | 11.61 | 20.87 | 257.11 | 12.32 |
| | W10 | MS5 | 23.51 | 292.00 | 12.42 | | | |
| | adw-A | | 20.67 | 260.05 | 12.58 | 19.98 | 265.81 | 13.30 |
| | adw-B | | 14.74 | 188.68 | 12.8 | | | |
| | adw-C | | 17.29 | 235.79 | 13.63 | 17.76 | 252.1 | 14.19 |
| The background tundra soils On Fildes Peninsula | GW1 | BS1 | 4.76 | 56.72 | 11.91 | 4.81 | 56.89 | 11.83 |
| | GW2 | BS2 | 5.05 | 56.63 | 11.21 | 5.2 | 63 | 12.12 |
| | GW3 | BS3 | 4.30 | 47.69 | 11.09 | | | |
| | gwc1 | | 3.29 | 31.78 | 9.66 | 3.1 | 35.4 | 11.42 |
| | gwc2 | | 3.09 | 29.65 | 9.6 | | | |
| | gwc3 | | 2.41 | 24.03 | 9.96 | 2.5 | 28.3 | 11.32 |
| | gwc4 | | 2.37 | 24.39 | 10.29 | | | |

*9. Line 219, "likewise, soil…", which site is author referring here? PTS or PLS? Or stating a generally pattern from PTS, PLS to MS? Please clarify. As PTS is clearly not showing this pattern.*

**Author response:** This only stated a general pattern from PS, PL sites to MS sites. Considering that PS sites do not show this pattern due to large spatial variations, this sentence was removed in the revised manuscript. The related description about soil chemical properties has been reorganized as follows:

PS and SS had generally lower C:N ratios than the penguin-lacking tundra soils (PL), tundra marsh soils (MS), and background tundra soils (BS). Soil mean TN, TS and $NH_4^+$–N levels were higher in PS, SS, PL, and MS than in BS. Soil $NH_4^+$–N contents were 1–2 orders of magnitude higher in PS and SS than in PL, MS, and BS, with the means of 176.9 and 137.6 mg $NH_4^+$-N $kg^{-1}$, respectively. The highest $NO_3^-$-N contents occurred in SS. Phosphorus levels were significantly greater ($p < 0.05$) in PS (10.6–32.9 mg $g^{-1}$) than in other types of tundra soils (mean < 6.0 mg $g^{-1}$). Overall, penguin or seal activities altered the local soil biogeochemical properties through the deposition of their excreta, leading to generally low C:N ratios. Please see the **line 226-236** in the revised manuscript.

*10. Line 222, "therefore, the soil TP and NH4…" this is a very bold statement, and lacking proof. Something like linear regression would be required.*

**Author response:** It is difficult to quantify **animal activity intensity,** therefore we do not use the phrase **"animal activity intensity"** to avoid ambiguity in the revised manuscript. This statement has been corrected as follows: **Overall, penguin or seal activities altered the local soil biogeochemical properties through the deposition of their excreta, leading to generally low C:N ratios in tundra soils.** Please see the **line 234-236** in the revised manuscript.

*11. Line 229, (fig. 2), figure 2 has 3 parts (a, b and c), please specify which part of the figure 2 is referred to.*

**Author response:** It is Fig. **2a**. This has been added in the revised manuscript (**line 247**).

*12. Line 229, "overall…" please provide a scatter plot to visualise this (can be put in supplementary)*

**Author response:** A scatter plot (**Fig. S2**) has been provided to visualize this in Supplementary Material.

*13. Line 231, "the archaeal amoA gene showed a heterogeneous distribution" what does heterogeneous distribution mean?*

**Author response:** It means that the AOA amoA gene showed a heterogeneous distribution **in the abundance** among the different tundra patches. i.e. The AOA amoA gene abundances were two orders of magnitude lower in PS and SS relative to those in BS and MS. The maximal AOA amoA gene abundance appeared in BS, followed by MS and PL, whereas the PS and SS soils had the lowest AOA amoA gene abundances. This sentence has been reorganized in the revised manuscript (**lines 249-253**).

*14. Line 232, there was a mixed usage of AOA amoA and archaeal amoA in the manuscript, please make them consistent.*

**Author response:** For consistency, we have used AOA *amoA* instead of archaeal

*amoA* in the revised manuscript.

*15. Line 237, "fig 3", similar to a previous comment, there are 6 parts of figure 3, please specify which part does this refer to.*

**Author response:** This figure related to sea animal activity intensity has been removed in the revised manuscript. We added **Fig. 3**: Effects of soil C:N alteration on AOA and AOB abundances, and potential ammonia oxidation rates (PAOR) at five tundra patches, to show effects of sea animal activities on AOA and AOB abundances and PAOR.

*16. Line 235, "Soil AOA amoA gene abundances were significant…" This statement is inappropriate, I would agree that animal activity reduces archaeal amoA gene abundance, but the statement of increasing archaeal gene abundance with reduced animal activity need a better proof. A correlation analysis between the activity intensity index and archaeal amoA gene abundance would be required.*

  **Author response:** Thanks for your good suggestion. This figure related to sea animal activity intensity has been removed in the revised manuscript, and the corresponding statement of increasing archaeal gene abundance with reduced animal activity has also been deleted. We added **Fig. 3**: Effects of soil C:N alteration on AOA and AOB abundances, and potential ammonia oxidation rates (PAOR) at five tundra patches, to show effects of sea animal activities on AOA and AOB abundances and PAOR. The related statements are reorganized as follows:

  **The log values of soil AOA *amoA* gene abundance showed a significant positive correlation (r=0.52, p<0.001) with C:N ratio (Fig. 3a), but a significant negative correlation with $NH_4^+$-N contents (r= -0.52, P = 0.013) (Table 2). Overall, penguin or seal activities, which were indicated by soil C:N ratios, significantly increased the abundance of soil AOB *amoA* genes, but reduced the abundance of AOA *amoA* genes, leading to very large ratios ($1.5 \times 10^2$ to $3.2 \times 10^4$) of AOB to AOA *amoA* copy numbers in PS and SS.** Please see **line 253-256, 270-274** in the revised manuscript.

*17. Line 240, "The soil AOB amoA gene abundances increased…" this is incorrect, author stated that the order of sampling reflected the intensity of seal activity (highest in SS1 and lowest in SS5) (line 123-127), but clearly the abundance of bacterial AOB gene reduced with reduced penguin or seal activity.*

  **Author response:** The related statements about animal activity intensity has been deleted. The related statements are reorganized as follows: **The log values of soil AOB**

*amoA* gene abundances showed a significant negative correlation with C:N ratio ($r$=-0.71, $P < 0.001$) (Fig. 3b), but significant positive correlation with $NH_4^+$-N ($r$=0.53 , $P < 0.05$) and TP ($r$=0.47 , $P < 0.05$) (Table 2). The ratios of AOB to AOA *amoA* copy numbers were strongly affected by animal activities, and were much higher in PS and SS than in PL, MS, and BS (Fig. 2b; Kruskal–Wallis test, $\chi^2 = 18.2$, $P = 0.01$). Their ratios showed significant positive correlation with $NH_4^+$-N contents ($r$=0.62; $P < 0.01$) and TP ($r$=0.43, $P < 0.05$) (Table 2), but significant negative correlation with the C:N ratios ($r$= -0.79; $P < 0.001$)(Fig. 3c). Overall, penguin or seal activities significantly increased the abundance of soil AOB *amoA* genes, but reduced the abundance of AOA *amoA* genes, leading to very large ratios ($1.5 \times 10^2$ to $3.2 \times 10^4$) of AOB to AOA *amoA* copy numbers in PS and SS. Please see **line 261-273** in the revised manuscript.

*18. line 242 "The ratios of AOB to AOA amoA…" please cite figure 2c for this sentence.*

    **Author response:** **Fig. 2b** has been cited for this sentence (**line 267**).

*19. line 250, " The PAOR was significantly higher in STS…", this is not fully correct, the PAOR of PS samples was not significantly different from the BS site.*

    **Author response:** This sentence has been reorganized as follows: The PAOR was slightly higher in SS (mean 76.1 μg N kg$^{-1}$ h$^{-1}$) than in PS (mean 64.7 μg N kg$^{-1}$ h$^{-1}$), but significantly higher than in PL, MS, and BS (mean 12.0–21.8 μg N kg$^{-1}$ h$^{-1}$). Overall the PAOR was significantly higher in animal colony soils (mean 70.4 μg N kg$^{-1}$ h$^{-1}$ for SS and PS) than in non-animal colony soils (15.7 μg N kg$^{-1}$ h$^{-1}$ for PL, MS, and BS; Kruskal–Wallis test, $\chi^2 = 11.6$, $P = 0.02$). Please see **line 277-281** in the revised manuscript.

*20. Figure 3, again, archaeal results appeared before bacterial results, thus their figure should appear before bacterial figures.*

    **Author response:** The figures for AOA results have been moved before the figures for AOB in the revised manuscript.

*21. Line 258, "Interestingly, the PAOR…" Please confirm this statement with a statistical analysis, as PAOR increased from SS3 to SS5.*

    **Author response:** This statement has been corrected and reorganized as follows: PAOR significantly negatively correlated with soil C:N ratios ($r$=0.73, $P<0.001$)(Fig. 3d), but significantly positively correlated with TS contents ($r$=0.47, $P<0.05$) and TP

contents (r=0.43, P<0.05) (Table 2). Please see line 290-293 in the revised manuscript.

*22. Line 271, "Specifically, the AOA amoA gene…" please present these results as a table or a figure.*

    **Author response:** These results have been provided in Table S1.

*23. Line 276 "Phylogenetic analysis showed that the AOA…" Why and how phylogenetic analysis was used to group sequences into OTUs? In addition, the entire sentence is confusing, please revise.*

    **Author response: (1)** The sequences with 97% identity were grouped into one OTU using the Mothur Program by the furthest neighbor approach (Zheng et al., 2014); **(2)** The entire sentence has been reorganized as: **Phylogenetic tree showed that the AOA *amoA* sequences were grouped into 16 unique OTUs, representing 100% of all the AOA *amoA* OTUs identified, and these sequences were affiliated with two *Nitrososphaera* clusters (Fig. 5a).** Please see line 311-313 in the revised manuscript.

*24. Line 289, "Phylogenetic analysis showed that AOB amoA…"Why and how phylogenetic analysis was used to group sequences into OTUs? In addition, the entire sentence is confusing, please revise.*

    **Author response: (1)** Phylogenetic analysis was used to find the evolutionary ties between species. The sequences were edited using DNAstar (DNASTAR, Madison, WI, USA), and then aligned by muscle using the UPGMB clustering method with the ClustalX program. The sequences with 97% identity were grouped into one OTU using the Mothur Program by the furthest neighbor approach. AOB *amoA* sequences were grouped into 65 unique OTUs in total, but the OTUs containing only one sequence were not displayed in the AOB phylogenetic tree. **(2)** The entire sentence has been revised as follows: **AOB phylogenetic tree showed 38 unique OTUs, representing 58.5% of all the AOB *amoA* OTUs identified, and they were grouped into four *Nitrosospira* clusters according to the evolutionary distance of the phylogenetic tree (Fig. 5b).** Please see line 345-347 in the revised manuscript.

*25. Line 312, "The AOA richness and phylotypes were evidently inhibited: : :" what does this mean? The richness of AOA is indeed lower in STS and PLS, but this result has already been presented in line 269.*

    **Author response:** It means the AOA richness was lower in SS and PL because of seal or penguin activities. This results has been presented on line 304-308, **therefore here this sentences was deleted in the revised manuscript.**

*26. Line 323 why RDA was used to investigate the correlation among amoA gene abundance, diversity and etc? I would think RDA is used to deal with matrix dataset, but all these variables are vector variable. If only correlations were required, Pearson or partial Pearson correlation would be sufficient. If the contribution of each variable is required, I would think VPA analysis would be a better option.*

    **Author response:** Thanks for your good suggestions. According to your suggestions, we deleted the description about the RDA analysis and results. Our data about environmental variables did not show normal distribution, therefore **we used Spearman correlation analysis to show their relationships between *amoA* gene abundance, the ratios of AOB to AOA, PAOR and environmental variables, and the results were given in Table 2.** Please see **line 255-256, line 263-264, line 268-269 and line 291-293** in the revised manuscript.

*27. Line 325, "The AOA amoA gene abundance: : :", which type of correlation is this? Please report the r value, and may be also scatter plots in the supplementary. Furthermore, authors stated that both AOA amoA gene abundance and diversity were related to C:N ratio, but only one P-value was reported.*

    **Author response: (1)** The description about the RDA analysis and results has been deleted in the revised manuscript, and Spearman correlation coefficients and P-values were given in the text and Table 2; **(2)** The scatter plots about *amoA* gene abundance, the ratios of AOB to AOA, PAOR and C:N ratios have been provided in Fig. 3.

*28. Table 1 need to provide full name of site, also the site codes do not match those in the main text.*

    **Author response:** The full name for the site has been given in Table 1, and all the site codes have been corrected for the consistency with the main text.

*29. Figure 2. The order of figure need to change, Figure 2b appeared first in the manuscript, and they should appear first in figure 2.*

    **Author response:** The order of this figure has been changed in the revised manuscript.

**Response to Referee #2**

At first, we would like to express our appreciation to the reviewer for your kind help and valuable comments about the revision of this manuscript (MS No.: bg-2019-114). We have considered your valuable suggestions and carefully revised this manuscript.

**The detailed responses inserted into reviewer #2 comments are attached as follows:**

*Anonymous Referee #2*

*The manuscript entitled "Effects of sea animal colonization on the coupling between dynamics and activity of soil ammonia-oxidizing bacteria and archaea in maritime Antarctica" by Wang et al. describes the effect of sea animal colonization on the community composition of ammonia oxidizers. The subject matter is interesting and the work in general is technically sound, however, my main concern is that the authors make claims about the relationship between nitrification rates and ammonia-oxidizer dynamics. Furthermore, there are some inconsistencies within the environmental parameter data, as well as very speculative parts in the discussion which need to be addressed.*

**Author response:** Thanks for your positive comments and valuable suggestions. We concentrated on *nitrification rates*, *some inconsistencies within the environmental parameter data, and speculative parts in the discussion,* and revised this manuscript carefully.

*General comments: The authors measured potential ammonia oxidation rates by adding 1mM NH$_4$Cl and incubating the samples at 15 degrees, which seems to be very artificial and far from in situ rates. It is highly speculative to comment on in-situ ammonia oxidation rates based in these measurements. Hence, assessing the relative contribution of AOA and AOB to nitrification rates based on the presented measurements is highly speculative and can only be suggested based on the differences in abundance between those two groups. Further, the authors talk about "inhibition" of AOA due to seal and penguin activities (e.g., lines 312-313, line 344), however, the presented data simply suggests a higher abundance of AOB over AOA. While the environmental conditions might be more favorable for AOB, it is highly speculative to assume that this is caused by inhibition and should be phrased more carefully.*

**Author response:** Thanks for your good comments. **(1)** Indeed we measured ammonia oxidation rates by adding 1mM $(NH_4)_2SO_4$ and incubating the samples at 15 °C, and they are different from in-situ ammonia oxidation rates. Therefore we used the word "**Potential** ammonia oxidation rates (PAOR)" to discriminate from "in-situ ammonia oxidation rates". **We concentrated the comparisons and analyses of POAR differences between the soils in tundra patches and their affecting factors.** The substrate concentration and incubation temperature in this study referred to several previous studies listed below.

| Sample | substrate concentration | incubation temperature | references |
| --- | --- | --- | --- |
| Antarctic soils | 1 mM $(NH_4)_2SO_4$ | room temperature | (Jung et al., 2011) |
| cold climate Soils | 1.25mM $(NH_4)_2SO_4$ | 25°C | (Fan et al., 2011) |
| Arctic soils | 1.7-2.5 mM $NH_4Cl$ | 15°C. | (Alves et al., 2013) |
| Antarctic soils | 1 mM $(NH_4)_2SO_4$ | 15°C. | This study |

Jung, J., Yeom, J., Kim, J., Han, J., Lim, H. S., Park, H., et al.: Change in gene abundance in the nitrogen biogeochemical cycle with temperature and nitrogen addition in Antarctic soils, *Research in Microbiology*, 162, 1018–1026, https://doi.org/10.1016/j.resmic.2011.07.007, 2011.

Fan, F., Yang, Q., Li, Z., Wei, D., Cui, X. A., and Liang, Y.: Impacts of organic and inorganic fertilizers on nitrification in a cold climate soil are linked to the bacterial ammonia oxidizer community, *Microbial Ecology*, 62, 982–990, https://doi.org/10.1007/s00248-011-9897-5, 2011.

Alves, R. J. E., Wanek, W., Zappe, A., Richter, A., Svenning, M. M., Schleper, C., and Urich, T.: Nitrification rates in Arctic soils are associated with functionally distinct populations of ammonia-oxidizing archaea, The *ISME* Journal, 7(8), 1620–1631, https://doi.org/10.1038/ ismej.2013.35, 2013.

**(2)** According to your comments, the relative contribution of AOA and AOB to nitrification rates was assessed based on the differences in abundance between the AOA and AOB groups and the correlation between their abundances and POAR; **(3)** The statement about "inhibition" has been removed, we phrased more carefully, and just say "the environmental conditions might be more favorable for AOB". Please see **line 382-385** and **line 414-417** in the revised manuscript with track changes.

*The ammonia concentrations of the 5 samples within the same site are sometimes extremely variable (e.g. 650 vs 0.1 in the STS site). How far were the different sampling points apart? Some of the data in Table 1 seems surprising or/and might be not well represented, e.g. the sum of the percentage of total carbon, nitrogen and sulfur makes up e.g. only 0.5%. What are the other 99.5%? Reporting total carbon, nitrogen and sulfur in mg/kg might be more useful as well. Additionally, the abbreviations of the sites are not very intuitive and easy to confuse.*

**Author response: (1)** In penguin or seal colonies, the penguin or seal populations showed high inhomogeneous distribution, and the deposition of penguin guano or seal excreta into the soil led to the large variations in soil TC, TN, TS, $NH_4^+$-N contents, even within very small tundra areas; **(2)** Our sampling points were 50-100 m apart. Soil nutrients N, P and S are higher in penguin or seal colony soils due to the deposition of penguin guano or seal excrements in maritime Antarctica. However, they are relatively lower in tundra areas moderately far away from animal colonies, and most of the soil are primary minerals, such as $SiO_2$, feldspar, mica and metallic oxides; **(3)** We used mg $g^{-1}$ to report total carbon, nitrogen and sulfur contents; **(4)** In the revised manuscript, we have used SS, PS, PL, MS, and BS for the samples and sites consistently to escape the ambiguity. **(5)** We have **rechecked** the measurement results of Table 1, and confirm that our data are right and valid. The reasons are as follows:

**a.** We measured soil TC and TN concentrations again, **which are provided in the following Table**, and the results are similar to those in this study, and their concentrations still showed large differences at the each sites within penguin or seal colony; **b.** We have measured the physiochemical properties of other soil samples in this area several times which were given in our previous published papers:

Zhu, R. B., Liu, Y. S., Xu, H., Ma, D. W., and Jiang, S.: Marine animals significantly increase tundra N2O and CH4 emissions in maritime Antarctica, *Journal of Geophysical Research: Biogeosciences*, 118(4), 1773–1792, https://doi.org/10.1002/2013JG002398, 2013.

Zhu RB, Liu YS, Ma ED, Sun JJ, Xu H, Sun LG. Nutrient compositions and potential greenhouse gas production in penguin guano, ornithogenic soils and seal colony soils in coastal Antarctica. *Antarctic Science*, doi:10.1017/S0954102009990204, 2009.

Soil chemical properties, especially $NH_4^+$-N, $NO_3^-$-N, P and S concentrations, also showed large differences due to effects of penguin or seal activities according to the two papers above.

Therefore we think that TC, TN, TS, TP, $NH_4^+$-N and $NO_3^-$-N levels showed high heterogeneity in penguin or seal colony tundra soils, PS and SS due to the deposition of penguin or seal excreta, and the differences of tundra vegetation and soil texture caused by animal tramp.

| | Original No. | No. in the paper | Detection in 2015 | | | Re-detection in 2019 | | |
|---|---|---|---|---|---|---|---|---|
| | | | N(mg/g) | C(mg/g) | C/N | N(mg/g) | C(mg/g) | C/N |
| Seal colony soils in western coast on Fildes Peninsula | SK1 | SS1 | 12.12 | 48.67 | 4.02 | 9.99 | 54.52 | 5.51 |
| | SK4 | SS2 | 16.94 | 70.06 | 4.13 | 13.38 | 81.81 | 6.15 |
| | SK6 | SS3 | 0.87 | 5.56 | 6.37 | 1.51 | 10.34 | 6.85 |
| | SK7 | | 2.40 | 13.64 | 5.69 | The sample is used up. | | |
| | SK8 | SS4 | 1.28 | 8.59 | 6.71 | The sample is used up. | | |
| | SK9 | | 2.63 | 18.88 | 7.19 | 2.51 | 18.98 | 7.56 |
| | SK10 | SS5 | 1.30 | 11.54 | 8.87 | The sample is used up, the same as below | | |
| Penguin colony soils on Ardley Island | E1 | | 10.54 | 50.58 | 4.8 | 8.68 | 55.83 | 6.43 |
| | E2 | PS1 | 14.55 | 84.65 | 5.82 | | | |
| | E3 | | 7.73 | 51.64 | 6.68 | 7.92 | 55.84 | 7.05 |
| | E4 | PS2 | 8.34 | 38.08 | 4.56 | | | |
| | E5 | | 15.07 | 89.71 | 5.95 | 13.48 | 92.33 | 6.85 |
| | E6 | PS3 | 17.90 | 120.76 | 6.75 | | | |
| | E7 | | 27.33 | 156.78 | 5.74 | 26.34 | 162.93 | 6.19 |
| | E8 | PS4 | 15.45 | 107.47 | 6.96 | | | |
| | E9 | | 9.99 | 73.10 | 7.31 | 8.87 | 79.72 | 8.99 |
| | E10 | PS5 | 7.97 | 45.82 | 5.75 | | | |
| The middle tundra soils on Ardley Island | M1 | PL1 | 11.53 | 117.64 | 10.2 | 9.88 | 124.91 | 12.64 |
| | M2 | | 13.61 | 138.41 | 10.17 | | | |
| | M3 | PL2 | 3.93 | 38.05 | 9.68 | 4.51 | 50.41 | 11.18 |
| | M4 | | 8.09 | 82.40 | 10.18 | | | |
| | M5 | PL3 | 25.30 | 302.52 | 11.96 | 23.94 | 301.93 | 12.61 |
| | M6 | | 20.19 | 222.45 | 11.02 | | | |
| | M7 | PL4 | 7.17 | 71.85 | 10.02 | 6.37 | 74.82 | 11.75 |
| | M8 | | 9.84 | 114.99 | 11.69 | | | |
| | M9 | | 11.47 | 110.65 | 9.65 | | | |
| | M10 | | 15.84 | 177.48 | 11.21 | 15.69 | 190.83 | 12.16 |
| | M11 | | 11.61 | 119.29 | 10.27 | | | |
| | M12 | | 4.34 | 44.40 | 10.23 | | | |

| | | | | | | | | |
|---|---|---|---|---|---|---|---|---|
| | M13 | | 9.65 | 116.36 | 12.05 | | | |
| | M14 | | 3.33 | 30.13 | 9.04 | 2.77 | 30.49 | 11.01 |
| | M15 | | 12.95 | 147.59 | 11.39 | | | |
| The tundra marsh soils in west of Ardley Island (almost no animals) | W1 | MS1 | 8.93 | 95.54 | 10.7 | 9.65 | 111.82 | 11.59 |
| | W2 | | 11.92 | 148.81 | 12.49 | | | |
| | W3 | MS2 | 15.89 | 193.95 | 12.2 | 14.35 | 191.57 | 13.35 |
| | W4 | | 17.83 | 217.76 | 12.21 | | | |
| | W5 | | 12.93 | 141.64 | 10.95 | 10.79 | 136.73 | 12.67 |
| | W6 | MS3 | 19.79 | 226.90 | 11.46 | | | |
| | W7 | | 10.81 | 122.84 | 11.37 | 9.37 | 122.43 | 13.07 |
| | W8 | MS4 | 26.57 | 355.02 | 13.36 | | | |
| | W9 | | 21.88 | 254.01 | 11.61 | 20.87 | 257.11 | 12.32 |
| | W10 | MS5 | 23.51 | 292.00 | 12.42 | | | |
| | adw-A | | 20.67 | 260.05 | 12.58 | 19.98 | 265.81 | 13.30 |
| | adw-B | | 14.74 | 188.68 | 12.8 | | | |
| | adw-C | | 17.29 | 235.79 | 13.63 | 17.76 | 252.1 | 14.19 |
| The background tundra soils On Fildes Peninsula | GW1 | BS1 | 4.76 | 56.72 | 11.91 | 4.81 | 56.89 | 11.83 |
| | GW2 | BS2 | 5.05 | 56.63 | 11.21 | 5.2 | 63 | 12.12 |
| | GW3 | BS3 | 4.30 | 47.69 | 11.09 | | | |
| | gwc1 | | 3.29 | 31.78 | 9.66 | 3.1 | 35.4 | 11.42 |
| | gwc2 | | 3.09 | 29.65 | 9.6 | | | |
| | gwc3 | | 2.41 | 24.03 | 9.96 | 2.5 | 28.3 | 11.32 |
| | gwc4 | | 2.37 | 24.39 | 10.29 | | | |

*Specific comments: Line 25: Nitrosospira are AOB and Nitrososphaera are AOA, needs to be switched.*

**Author response:** The order has been switched in this sentence. Please see **line 26-27** in revised manuscript.

*Line 32-33: "The results provide insights into the mechanism how microbes drive nitrification in maritime Antarctica", here again the authors make claims that are not supported by the presented data. The mechanisms of nitrification are not studied.*

**Author response:** According to your suggestion, this sentence has been removed in the revised manuscript.

*Line 37: "biogeochemical nitrogen cycle" instead of "biogeochemical cycle for nitrogen"*

**Author response:** This has been corrected in the revised manuscript. Please see **line 40** in revised manuscript.

*Line 40: AOB were discovered much earlier than 2015, please chose a different reference*

**Author response:** The reference has been changed as follows:

Belser, L. W., and Schmidt, E. L. Diversity in the ammonia-oxidizing nitrifier population of a soil. Applied and Environmental Microbiology, 36, 584–588, 1978.

Please see **line 43** in revised manuscript.

*Line 41: comammox should be spelled out*

**Author response:** According to another reviewer's comments, this sentence and *comammox* is out of picture. Therefore this sentence and *comammox* have been removed in the revised manuscript.

*Line 46: Are you referring to the marine water column or sediments? Please specify (instead of mentioning "marine layers") and add the appropriate references.*

**Author response:** The research object of Baker et al (2012) and Bouskill et al (2012) was marine water column. "Oxic and suboxic marine layers" has been replaced by "**oxic and suboxic marine water column**" for more accurate expression. The references "Baker et al (2012) and Bouskill et al (2012)" are still used in the revised manuscript. Please see **line 49-50** in revised manuscript.

*Line 93: "daily mean range" is contradictory, please correct.*

**Author response:** This sentence is to explain that the minimum daily mean temperature is -22.6℃ in winter, and the highest daily mean temperature is 11.7 ℃, which occurs in summer. This has been corrected into "**the range of daily mean temperature**". Please see **line 97** in revised manuscript.

*Line 101: "A great many" should probably read "A great majority"*

**Author response:** This has been corrected into "**A great majority**". Please see **line 105** in revised manuscript.

*Line 346: typo in "reported"*

**Author response: This has been corrected in the revised manuscript (line 386).**

*Lines 378-380: This statement is not necessarily correct. There might be more diversity within Km's of AOA that differ from that of N. maritimus. Making such a claim based on a single organism is very speculative.*

**Author response:** I agree with your comments, AOA group I.1b **might** exhibit a broader range of metabolism and adaptation and making such a claim **based on a single organism** is very speculative. We have removed this statement "……because the half-saturation constant for ammonia oxidation by *Thaumarchaeota* is lower than that by AOB", and only discussed the effects of $NH_4^+$-N levels on the AOA abundance and diversity, based on the correlation between $NH_4^+$-N levels on the AOA abundance and cited more references (Stieglmeier et al., 2014): This statement is reorganized as follows:

**The AOA abundance showed a significant negative correlation with $NH_4^+$-N levels in tundra patches (Table 2), indicating that AOA might better adapt to low $NH_4^+$ and oligotrophic environments (Martens-Habbena et al., 2009; Stieglmeier et al., 2014). High $NH_4^+$-N concentrations might partially inhibit AOA populations (Hatzenpichler et al., 2008). This result is similar to that reported for some agricultural soils with increased fertilization, and grassland soils with increased grazing (Fan et al., 2011; Prosser and Nicol, 2012; Pan et al., 2018), supporting the conclusion that AOA and AOB generally inhabit different niches in soil, distinguished by the $NH_4^+$ concentration and availability (Verhamme et al., 2011; Wessén et al., 2011).** Please see **line 420-431** in revised manuscript.

*Lines 393-397: The connection with comammox is not very intuitive. Did you detect comammox? Also, the reference of Santoro 2016 does not fit here because it measures actual rates (instead of potential rates) using stable isotopes in marine environments where no comammox has been found thus far.*

**Author response:** According to your comments, this statement and the reference have been removed in the revised manuscript.

*Lines 417-420: Why does a high organic carbon favor AOA over AOB? So far most studies have shown that AOA are inhibited by complex organic substrates (Stieglmeier et al 2011, Qin et al 2017, etc).*

**Author response:** This statement has been corrected and reorganized as follows: The BS and MS were moderately far away from penguin or seal colonies without the input of the nutrients from sea animal excrements, and their substrates can be provided only through the mineralization of organic matter from local tundra plants. The simple organic substrates and barren soil environment might favor AOA (Stopnišeket al., 2010; Habteselassie et al., 2013). Therefore, AOA showed relatively high abundance in MS and BS compared with PS and SS. Please see line 467-474 in revised manuscript.

*Lines 430-433: This statement is highly speculative and likely wrong. Why would the presence of an amoA gene be an ancestral remnant that is not active? There is no data presented supporting such claims.*

**Author response:** Thanks. I agree with you. According to your comments, we have removed this statement.

*Lines 446-455: this section does not discuss the data and should be moved to results*

**Author response:** This section has been delete, considering that the information was already provided in Figure 5b.

Effects of Sea Animal Colonization on the Coupling between Dynamics and

Activity of Soil Ammonia-oxidizing Bacteria and Archaea in Maritime Antarctica

Qing Wang [1], Renbin Zhu[1,]*, Yanling Zheng[2], Tao Bao[1], Lijun Hou[2,]*

[1]Anhui Province Key Laboratory of Polar Environment and Global Change, School of Earth and

Space Sciences, University of Science and Technology of China, Hefei 230026, P.R China

[2]State Key Laboratory of Estuarine and Coastal Research, East China Normal University,

Shanghai 200062, P. R China

*Corresponding author: Renbin Zhu (zhurb@ustc.edu.cn) or Lijun Hou (ljhou@sklec.ecnu.edu.cn)

**Abstract**

The colonization  by a large number of sea animal, including penguins and seals, plays an important role in the nitrogen cycle of the tundra ecosystem in coastal Antarctica. However, little is known about the effects of sea animal colonization on ammonia-oxidizing archaea (AOA) and bacteria (AOB) communities involved in nitrogen transformations. In this study, we chose active seal colony tundra soils (S~T~S), penguin colony soils (P~T~S), adjacent penguin-lacking tundra soils (PL~S~), 
[revised manuscript text omitted]